# Robustness via Uncertainty-aware Cycle Consistency

**Uddeshya Upadhyay**[1]     **Yanbei Chen**[1]     **Zeynep Akata**[1,2]

[1]University of Tübingen     [2]Max Planck Institute for Intelligent Systems

## Abstract

Unpaired image-to-image translation refers to learning inter-image-domain mapping without corresponding image pairs. Existing methods learn deterministic mappings without explicitly modelling the robustness to outliers or predictive uncertainty, leading to performance degradation when encountering unseen perturbations at test time. To address this, we propose a novel probabilistic method based on Uncertainty-aware Generalized Adaptive Cycle Consistency (UGAC), which models the per-pixel residual by generalized Gaussian distribution, capable of modelling heavy-tailed distributions. We compare our model with a wide variety of state-of-the-art methods on various challenging tasks including unpaired image translation of natural images, using standard datasets, spanning autonomous driving, maps, facades, and also in medical imaging domain consisting of MRI. Experimental results demonstrate that our method exhibits stronger robustness towards unseen perturbations in test data. Code is released here: https://github.com/ExplainableML/UncertaintyAwareCycleConsistency.

## 1  Introduction

Translating an image from a distribution, i.e. source domain, to an image in another distribution, i.e. target domain, with a distribution shift is an ill-posed problem as a unique deterministic one-to-one mapping may not exist between the two domains. Furthermore, since the correspondence between inter-domain samples may be missing, their joint-distribution needs to be inferred from a set of marginal distributions. However, as infinitely many joint distributions can be decomposed into a fixed set of marginal distributions [1, 2, 3], the problem is ill-posed in the absence of additional constraints.

Deep learning-based methods tackle the image-to-image translation task by learning inter-domain mappings in a paired or unpaired manner. Paired image translation methods [4, 5, 6, 7, 8, 9] exploit the inter-domain correspondence by penalizing the per-pixel residual (using $l_1$ or $l_2$ norm) between the output and corresponding ground-truth sample. Unpaired image translation approaches [1, 10, 11, 12, 13, 14] often use adversarial networks with an additional constraint on the image or feature space imposing structure on the underlying joint distribution of the images from the different domains.

Both paired and unpaired image translation approaches often learn a deterministic mapping between the domains where every pixel in the input domain is mapped to a fixed pixel value in the output domain. However, such a deterministic formulation can lead to mode collapse while at the same time not being able to quantify the model predictive uncertainty important for critical applications, e.g., medical image analysis. It is desirable to test the performance of the model on unseen perturbed input at test-time, to improve their applicability in the real world. While robustness to outliers is a focus in some domains [15, 16, 17, 18], it has not attracted as much attention in unpaired translation.

To address these limitations, we propose an unpaired (unsupervised) probabilistic image-to-image translation method trained without inter-domain correspondence in an end-to-end manner. The probabilistic nature of this method provides uncertainty estimates for the predictions. Moreover, modelling the residuals between the predictions and the ground-truth with heavy-tailed distributions

35th Conference on Neural Information Processing Systems (NeurIPS 2021).

makes our model robust to outliers and various unseen data. Accordingly, we compare various state-of-the-art models and our model in their capacity to handle samples from similar distribution as training-dataset as well as perturbed samples, in the context of unpaired translation.

Our contributions are as follows. (i) We propose an *unpaired* probabilistic image-to-image translation framework based on Uncertainty-aware Generalized Adaptive Cycle Consistency (UGAC). Our framework models the residuals between the predictions and the ground-truths with heavy-tailed distributions improving robustness to outliers. Probabilistic nature of UGAC also provides uncertainty estimates for the predictions. (ii) We evaluate UGAC on multiple challenging datasets: natural images consisting Cityscapes [19], Google aerial maps and photos [4], CMP Facade [20] and medical images consisting of MRI from IXI [21]. We compare our model to seven state-of-the-art image-to-image translation methods [12, 22, 1, 11, 10, 23]. Our results demonstrate that while UGAC performs competitively when tested on unperturbed images, it improves state-of-the-art methods substantially when tested on unseen perturbations, establishing its robustness. (iii) We show that our estimated uncertainty scores correlate with the model predictive errors (i.e., residual between model prediction and the ground-truth) suggesting that it acts as a good proxy for the model's reliability at test time.

## 2  Related Work

**Image-to-image translation.** Image-to-image translation is often formulated as per-pixel deterministic regression between two image domains of [24, 25, 26]. In [4], this is done in a *paired* manner using conditional adversarial networks, while in [10, 1, 12, 22, 11] this is done in an *unpaired* manner by enforcing additional constraints on the joint distribution of the images from separate domains. Both CycleGAN [10] and UNIT [1] learn bi-directional mappings, whereas other recent methods [12, 22, 11] learn uni-directional mappings.

Quantification of uncertainty in the predictions made by the unpaired image-to-image translation models largely remains unexplored. Our proposed method operates at the intersection of uncertainty estimation and unsupervised translation. Critical applications such as medical image-to-image translation [27, 28, 29, 30, 31, 32] is an excellent testbed for our model as confidence in the network's predictions is desirable [33, 34] especially under the influence of missing imaging modalities.

**Uncertainty estimation.** Among two broad categories of uncertainties that can be associated with a model's prediction, *epistemic* uncertainty in the model parameters is learned with finite data whereas *aleatoric* uncertainty captures the noise/uncertainty inherent in the data [35, 36]. For image-to-image translation, various uncertainties can be estimated using Bayesian deep learning techniques [36, 37, 38, 39, 40]. In critical areas like medical imaging, the errors in the predictions deter the adoption of such frameworks in clinical contexts. Uncertainty estimates for the predictions would allow subsequent revision by clinicians [41, 42, 43, 44, 45, 46, 47, 48, 49, 50, 51].

Existing methods model the per-pixel *heteroscedasticity* as Gaussian distribution for regression tasks [36]. This is not optimal in the presence of outliers that often tend to follow heavy-tailed distributions [52, 53]. Therefore, we enhance the above setup by modelling per-pixel heteroscedasticity as generalized Gaussian distribution, which can model a wide variety of distributions, including Gaussian, Laplace, and heavier-tailed distribution.

## 3  Uncertainty-aware Generalized Adaptive Cycle-consistency (UGAC)

We present the formulation of the unpaired image-to-image translation problem. We discuss the shortcomings of the existing solution involving the cycle consistency loss called CycleGAN [10]. Finally, we present our novel probabilistic framework (UGAC) that overcomes the described shortcomings.

### 3.1  Preliminaries

**Formulation.** Let there be two image domains $A$ and $B$. Let the set of images from domain $A$ and $B$ be defined by (i) $S_A := \{a_1, a_2...a_n\}$, where $a_i \sim \mathcal{P}_A \; \forall i$ and (ii) $S_B := \{b_1, b_2...b_m\}$, where $b_i \sim \mathcal{P}_B \; \forall i$, respectively. The elements $a_i$ and $b_i$ represent the $i^{th}$ image from domain $A$ and $B$ respectively, and are drawn from an underlying *unknown* probability distribution $\mathcal{P}_A$ and $\mathcal{P}_B$ respectively.

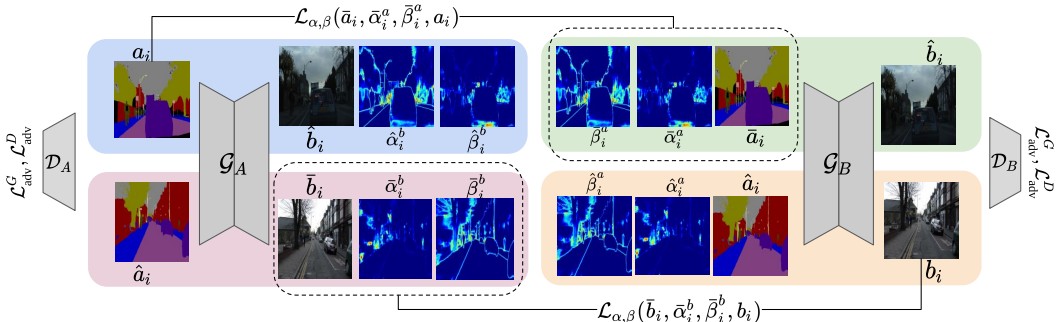

Figure 1: Our UGAC framework with the cycle between two generators. For translating from $A$ to $B$ ($A \rightarrow B$), the input $a_i$ is mapped to generalized Gaussian distribution parameterized by $\{\hat{b}_i, \hat{\alpha}_i^b, \hat{\beta}_i^b\}$. The backward cycle ($A \rightarrow B \rightarrow A$) reconstructs the image distribution parameterized by $\{\bar{a}_i, \bar{\alpha}_i^a, \bar{\beta}_i^a\}$. UGAC uses $\mathcal{L}_{\alpha\beta}$ objective function in Eq. 8 and adversarial losses in Eq. 10 and 11.

Let each image have $K$ pixels, and $u_{ik}$ represent the $k^{th}$ pixel of a particular image $u_i$. We are interested in learning a mapping from domain $A$ to $B$ ($A \rightarrow B$) and $B$ to $A$ ($B \rightarrow A$) in an unpaired manner so that the correspondence between the samples from $\mathcal{P}_A$ and $\mathcal{P}_B$ is not required at the learning stage. In other words, we want to learn the underlying joint distribution $\mathcal{P}_{AB}$ from the given marginal distributions $\mathcal{P}_A$ and $\mathcal{P}_B$. This work utilizes CycleGANs that leverage the cycle consistency to learn mappings from both directions ($A \rightarrow B$ and $B \rightarrow A$), but often we are only interested in one direction and the second direction is the auxiliary mapping that aids in learning process. We define the mapping $A \rightarrow B$ as primary and $B \rightarrow A$ as auxiliary.

**Cycle consistency.** Learning a joint distribution from the marginal distributions is an ill-posed problem with infinitely many solutions [3]. CycleGAN [10] enforces an additional structure on the joint distribution using a set of primary networks (forming a GAN) and a set of auxiliary networks. The primary networks are represented by $\{\mathcal{G}_A(\cdot; \theta_A^{\mathcal{G}}), \mathcal{D}_A(\cdot; \theta_A^{\mathcal{D}})\}$, where $\mathcal{G}_A$ represents a generator and $\mathcal{D}_A$ represents a discriminator. The auxiliary networks are represented by $\{\mathcal{G}_B(\cdot; \theta_B^{\mathcal{G}}), \mathcal{D}_B(\cdot; \theta_B^{\mathcal{D}})\}$. While the primary networks learn the mapping $A \rightarrow B$, the auxiliary networks learn $B \rightarrow A$ (see Figure 1). Let the output of the generator $\mathcal{G}_A$ translating samples from domain $A$ (say $a_i$) to domain $B$ be called $\hat{b}_i$. Similarly, for the generator $\mathcal{G}_B$ translating samples from domain $B$ (say $b_i$) to domain $A$ be called $\hat{a}_i$, i.e., $\hat{b}_i = \mathcal{G}_A(a_i; \theta_A^{\mathcal{G}})$ and $\hat{a}_i = \mathcal{G}_B(b_i; \theta_B^{\mathcal{G}})$. To simplify the notation, we will omit writing parameters of the networks in the equation. The cycle consistency constraint [10] re-translates the above predictions ($\hat{b}_i, \hat{a}_i$) to get back the reconstruction in the original domain ($\bar{a}_i, \bar{b}_i$), where, $\bar{a}_i = \mathcal{G}_B(\hat{b}_i)$ and $\bar{b}_i = \mathcal{G}_A(\hat{a}_i)$, and attempts to make reconstructed images ($\bar{a}_i, \bar{b}_i$) similar to original input ($a_i, b_i$) by penalizing the residuals with $\mathcal{L}_1$ norm between the reconstructions and the original input images, giving the cycle consistency $\mathcal{L}_{\text{cyc}}(\bar{a}_i, \bar{b}_i, a_i, b_i) = \mathcal{L}_1(\bar{a}_i, a_i) + \mathcal{L}_1(\bar{b}_i, b_i)$.

**Limitations of cycle consistency.** The underlying assumption when penalizing with the $\mathcal{L}_1$ norm is that the residual at *every pixel* between the reconstruction and the input follow *zero-mean and fixed-variance Laplace* distribution, i.e., $\bar{a}_{ij} = a_{ij} + \epsilon_{ij}^a$ and $\bar{b}_{ij} = b_{ij} + \epsilon_{ij}^b$ with,

$$\epsilon_{ij}^a, \epsilon_{ij}^b \sim Laplace(\epsilon; 0, \frac{\sigma}{\sqrt{2}}) \equiv \frac{1}{\sqrt{2\sigma^2}} e^{-\sqrt{2}\frac{|\epsilon - 0|}{\sigma}}, \tag{1}$$

where $\sigma^2$ represents the fixed-variance of the distribution, $a_{ij}$ represents the $j^{th}$ pixel in image $a_i$, and $\epsilon_{ij}^a$ represents the noise in the $j^{th}$ pixel for the estimated image $\bar{a}_{ij}$. This assumption on the residuals between the reconstruction and the input enforces the likelihood (i.e., $\mathscr{L}(\Theta|\mathcal{X}) = \mathcal{P}(\mathcal{X}|\Theta)$, where $\Theta := \theta_A^{\mathcal{G}} \cup \theta_B^{\mathcal{G}} \cup \theta_A^{\mathcal{D}} \cup \theta_B^{\mathcal{D}}$ and $\mathcal{X} := S_A \cup S_B$) to follow a *factored Laplace* distribution:

$$\mathscr{L}(\Theta|\mathcal{X}) \propto \prod_{ijpq} e^{-\frac{\sqrt{2}|\bar{a}_{ij} - a_{ij}|}{\sigma}} e^{-\frac{\sqrt{2}|\bar{b}_{pq} - b_{pq}|}{\sigma}}, \tag{2}$$

where minimizing the negative-log-likelihood yields $\mathcal{L}_{\text{cyc}}$ with the following limitations. The residuals in the presence of outliers may not follow the Laplace distribution but instead a heavy-tailed distribution, whereas the i.i.d assumption leads to fixed variance distributions for the residuals that do not allow modelling of *heteroscedasticity* to aid in uncertainty estimation.

## 3.2 Building Uncertainty-aware Cycle Consistency

We propose to alleviate the mentioned issues by modelling the underlying per-pixel residual distribution as independent but *non-identically* distributed *zero-mean generalized Gaussian distribution* (GGD) (Figure 2), i.e., with no fixed shape ($\beta > 0$) and scale ($\alpha > 0$) parameters. Instead, all the shape and scale parameters of the distributions are predicted from the networks and formulated as follows:

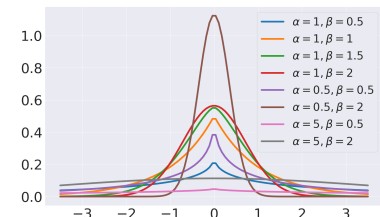

Figure 2: Probability density function (pdf) for generalized Gaussian distribution. Different scale ($\alpha$) and shape ($\beta$) parameters lead to different tail behaviour. $(\alpha, \beta) = (1, 2)$ represents Gaussian distribution.

$$\epsilon_{ij}^a, \epsilon_{ij}^b \sim GGD(\epsilon; 0, \bar{\alpha}_{ij}, \bar{\beta}_{ij}) \equiv \frac{\bar{\beta}_{ij}}{2\bar{\alpha}_{ij}\Gamma(\frac{1}{\bar{\beta}_{ij}})} e^{-\left(\frac{|\epsilon - 0|}{\bar{\alpha}_{ij}}\right)^{\bar{\beta}_{ij}}}. \tag{3}$$

For each $\epsilon_{ij}$, the parameters of the distribution $\{\bar{\alpha}_{ij}, \bar{\beta}_{ij}\}$ may not be the same as parameters for other $\epsilon_{ik}$s; therefore, they are non-identically distributed allowing modelling with heavier tail distributions. The likelihood for our proposed model is,

$$\mathcal{L}(\Theta|\mathcal{X}) = \prod_{ijpq} \mathscr{G}(\bar{\beta}_{ij}^a, \bar{\alpha}_{ij}^a, \bar{a}_{ij}, a_{ij}) \mathscr{G}(\bar{\beta}_{pq}^b, \bar{\alpha}_{pq}^b, \bar{b}_{pq}, b_{pq}), \tag{4}$$

where $(\bar{\beta}_{ij}^a)$ represents the $j^{th}$ pixel of domain $A$'s shape parameter $\beta_i^a$ (similarly for others). $\mathscr{G}(\bar{\beta}_{ij}^u, \bar{\alpha}_{ij}^u, \bar{u}_{ij}, u_{ij})$ is the pixel-likelihood at $j^{th}$ pixel of image $u_i$ (that can represent images of both domain $A$ and $B$) formulated as,

$$\mathscr{G}(\bar{\beta}_{ij}^u, \bar{\alpha}_{ij}^u, \bar{u}_{ij}, u_{ij}) = GGD(u_{ij}; \bar{u}_{ij}, \bar{\alpha}_{ij}^u, \bar{\beta}_{ij}^u), \tag{5}$$

The negative-log-likelihood is given by,

$$-\ln \mathcal{L}(\Theta|\mathcal{X}) = -\sum_{ijpq} \left[ \ln \frac{\bar{\beta}_{ij}^a}{2\bar{\alpha}_{ij}^a\Gamma(\frac{1}{\bar{\beta}_{ij}^a})} e^{-\left(\frac{|\bar{a}_{ij} - a_{ij}|}{\bar{\alpha}_{ij}^a}\right)^{\bar{\beta}_{ij}^a}} + \ln \frac{\bar{\beta}_{pq}^b}{2\bar{\alpha}_{pq}^b\Gamma(\frac{1}{\bar{\beta}_{pq}^b})} e^{-\left(\frac{|\bar{b}_{pq} - b_{pq}|}{\bar{\alpha}_{pq}^b}\right)^{\bar{\beta}_{pq}^b}} \right] \tag{6}$$

minimizing the negative-log-likelihood yields a new cycle consistency loss, which we call as the uncertainty-aware generalized adaptive cycle consistency loss $\mathcal{L}_{ucyc}$, given $\mathscr{A} = \{\bar{a}_i, \bar{\alpha}_i^a, \bar{\beta}_i^a, a_i\}$ and $\mathscr{B} = \{\bar{b}_i, \bar{\alpha}_i^b, \bar{\beta}_i^b, b_i\}$,

$$\mathcal{L}_{ucyc}(\mathscr{A}, \mathscr{B}) = \mathcal{L}_{\alpha\beta}(\mathscr{A}) + \mathcal{L}_{\alpha\beta}(\mathscr{B}), \tag{7}$$

where $\mathcal{L}_{\alpha\beta}(\mathscr{A}) = \mathcal{L}_{\alpha\beta}(\bar{a}_i, \bar{\alpha}_i^a, \bar{\beta}_i^a, a_i)$ is the new objective function corresponding to domain $A$,

$$\mathcal{L}_{\alpha\beta}(\bar{a}_i, \bar{\alpha}_i^a, \bar{\beta}_i^a, a_i) = \frac{1}{K} \sum_j \left( \frac{|\bar{a}_{ij} - a_{ij}|}{\bar{\alpha}_{ij}^a} \right)^{\bar{\beta}_{ij}^a} - \log \frac{\bar{\beta}_{ij}^a}{\bar{\alpha}_{ij}^a} + \log \Gamma(\frac{1}{\bar{\beta}_{ij}^a}), \tag{8}$$

where $(\bar{a}_i, \bar{b}_i)$ are the reconstructions for $(a_i, b_i)$ and $(\bar{\alpha}_i^a, \bar{\beta}_i^a)$, $(\bar{\alpha}_i^b, \bar{\beta}_i^b)$ are scale and shape parameters for the reconstruction $(\bar{a}_i, \bar{b}_i)$, respectively.

The $\mathcal{L}_1$ norm-based cycle consistency ($\mathcal{L}_{cyc}$) is a special case of $\mathcal{L}_{ucyc}$ with $(\bar{\alpha}_{ij}^a, \bar{\beta}_{ij}^a, \bar{\alpha}_{ij}^b, \bar{\beta}_{ij}^b) = (1, 1, 1, 1) \forall i, j$. To utilize $\mathcal{L}_{ucyc}$, one must have the $\alpha$ maps and the $\beta$ maps for the reconstructions of the inputs. To obtain the reconstructed image, $\alpha$ (scale map), and $\beta$ (shape map), we modify the head of the generators (the last few convolutional layers) and split them into three heads, connected to a common backbone. Therefore, for inputs $a_i$ and $b_i$ to the generator $\mathcal{G}_A$ and $\mathcal{G}_B$, the outputs are:

$$(\hat{b}_i, \hat{\alpha}_i^b, \hat{\beta}_i^b) = \mathcal{G}_A(a_i) \text{ and } (\bar{a}_i, \bar{\alpha}_i^a, \bar{\beta}_i^a) = \mathcal{G}_B(\hat{b}_i)$$
$$(\hat{a}_i, \hat{\alpha}_i^a, \hat{\beta}_i^a) = \mathcal{G}_B(b_i) \text{ and } (\bar{b}_i, \bar{\alpha}_i^b, \bar{\beta}_i^b) = \mathcal{G}_A(\hat{a}_i), \tag{9}$$

The estimates are plugged into Eq. (7) and the networks are trained to estimate all the parameters of the GGD modelling domain $A$ and $B$, i.e. $(\bar{a}_{ij}, \bar{\alpha}_{ij}^a, \bar{\beta}_{ij}^a)$ and $(\bar{b}_{ij}, \bar{\alpha}_{ij}^b, \bar{\beta}_{ij}^b) \forall ij$.

Furthermore, we apply adversarial losses [10] to the mapping functions, (i) $\mathcal{G}_A : A \rightarrow B$ and (ii) $\mathcal{G}_B : B \rightarrow A$, using the discriminators $\mathcal{D}_A$ and $\mathcal{D}_B$. The discriminators are inspired from patchGANs [4, 10] that classify whether 70x70 overlapping patches are real or not. The adversarial loss for the generators ($\mathcal{L}_{\text{adv}}^G$ [10]) is,

$$\mathcal{L}_{\text{adv}}^G = \mathcal{L}_2(\mathcal{D}^A(\hat{b}_i), 1) + \mathcal{L}_2(\mathcal{D}^B(\hat{a}_i), 1). \tag{10}$$

The loss for discriminators ($\mathcal{L}_{\text{adv}}^D$ [10]) is,

$$\mathcal{L}_{\text{adv}}^D = \mathcal{L}_2(\mathcal{D}^A(b_i), 1) + \mathcal{L}_2(\mathcal{D}^A(\hat{b}_i), 0) + \mathcal{L}_2(\mathcal{D}^B(a_i), 1) + \mathcal{L}_2(\mathcal{D}^B(\hat{a}_i), 0). \tag{11}$$

To train the networks we update the generator and discriminator sequentially at every step [10, 4, 54]. The generators and discriminators are trained to minimize $\mathcal{L}^G$ and $\mathcal{L}^D$ as follows:

$$\mathcal{L}^G = \lambda_1 \mathcal{L}_{\text{ucyc}} + \lambda_2 \mathcal{L}_{\text{adv}}^G \text{ and } \mathcal{L}^D = \mathcal{L}_{\text{adv}}^D. \tag{12}$$

**Closed-form solution for aleatoric uncertainty.** Although predicting parameters of the output image distribution allows to sample multiple images for the same input and compute the uncertainty, modelling the distribution as GGD gives us the uncertainty ($\sigma_{\text{aleatoric}}$) without sampling from the distribution as a closed form solution exists, $\sigma_{\text{aleatoric}}^2 = \frac{\alpha^2 \Gamma(\frac{3}{\beta})}{\Gamma(\frac{1}{\beta})}$. Epistemic uncertainty ($\sigma_{\text{epistemic}}$) is calculated by multiple forward passes ($T = 50$ times) with dropouts activated for the same input and computing the variance across the outputs ($\hat{u}_t$), i.e., $\sigma_{\text{epistemic}}^2 = (\sum_t (\hat{u}_t - \sum_t \frac{\hat{u}_t}{T})^2)/T$. We define the total uncertainty ($\sigma$) as $\sigma^2 = \sigma_{\text{aleatoric}}^2 + \sigma_{\text{epistemic}}^2$.

## 4 Experiments

In this section, we first describe our experimental setup and implementation details. We compare our model to a wide variety of state-of-the-art methods quantitatively and qualitatively. Finally we provide an ablation analysis to study the rationale of our model formulation.

### 4.1 Experimental Setup

**Tasks.** We study the robustness of unpaired image-to-image translation methods, where different methods are first trained on *clean* images and then evaluated on *perturbed* images The *clean* images are referred as noise-level 0 (NL0); while the *perturbed* images with *increasing* noise are referred as NL1, NL2, and NL3. We test three types of perturbation including Gaussian, Uniform, and Impulse. From NL0 to NL3, the standard deviation of the additive Gaussian noise is gradually increased. Similarly, for additive uniform noise, different levels are obtained by gradually increase the upper-bound of the uniform sampling interval [55] and for impulse noise we gradually increase the probability of pixel-value replacement [56]. Details of constructing various NLs are in supplementary.

**Datasets.** We evaluate on four standard datasets used for image-to-image translation: (i) *Cityscapes* [19] contains street scene images with segmentation maps, including 2,975 training and 500 validation and test images; (ii) *Google maps* [4] contains 1,096 training and test images scraped from Google maps with aerial photographs and maps; (iii) *CMP Facade* [20] contains 400 images from the CMP Facade Database including architectural facades labels and photos. (iv) *IXI* [21] is a medical imaging dataset with 15,000/5,000/10,000 training/test/validation images, including T1 MRI and T2 MRI. More preprocessing details for all the datasets are in supplementary.

**Translation quality metrics.** Following [23], we evaluate the translation quality of the generated segmentation maps and images, for the datasets with segmentation maps (e.g., Cityscapes). First, to evaluate the generated segmentation maps, we compute the Intersection over union (`IoU.SEGM`) and mean class-wise accuracy (`Acc.SEGM`) between the generated segmentation maps and the ground-truth segmentation maps. Second, to evaluate the generated images, we first feed the generated images $X_{tr}$ to a pre-trained pix2pix model [4] (denoted as $p2p$, which is trained to translate images to segmentation maps) to obtain the segmentation maps $p2p(X_{tr})$. Then, we feed the original images $X_{org}$ to the same pix2pix model to obtain another segmentation maps $p2p(X_{org})$, and compute the IoU between two outputs $p2p(X_{tr})$ and $p2p(X_{org})$ (`IoU.P2P`).

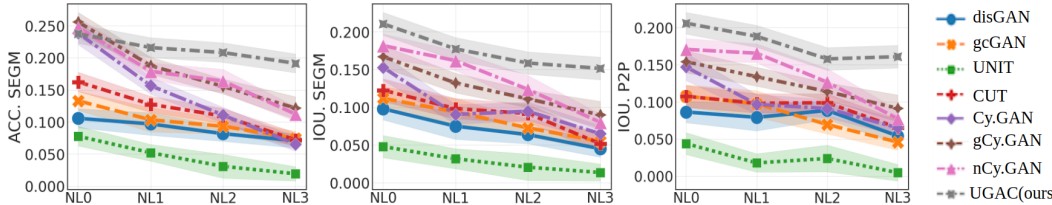

Figure 3: Evaluation of different methods on Cityscapes with Gaussian perturbation under varying noise levels. NL0 denotes clean images without noise, NL1, NL2, NL3 are unseen noise levels. `ACC.Segm`, `IoU.Segm`, `IoU.P2P` are three metrics for evaluating translation quality. Higher is better.

| P | Methods | Cityscapes | | Maps | | Facade | | IXI | |
|---|---------|-----------|-----------|-----------|-----------|-----------|-----------|-----------|-----------|
| | | AMSE (std)↓ | ASSIM (std)↑ | AMSE (std)↓ | ASSIM (std)↑ | AMSE (std)↓ | ASSIM (std)↑ | AMSE (std)↓ | ASSIM (std)↑ |
| Gaussian | gcGAN [22] | 107.83 (10.8) | 0.62 (0.09) | 117.21 (10.6) | 0.43 (0.07) | 138.21 (11.5) | 0.41 (0.05) | 108.32 (8.7) | 0.67 (0.12) |
| | CUT [11] | 108.34 (8.7) | 0.51 (0.12) | 119.32 (8.9) | 0.51 (0.11) | 123.22 (17.6) | 0.58 (0.09) | 87.12 (10.4) | 0.64 (0.07) |
| | Cy.GAN [10] | 121.32 (10.3) | 0.31 (0.13) | 107.32 (7.5) | 0.61 (0.13) | 134.23 (15.3) | 0.45 (0.07) | 98.14 (9.1) | 0.70 (0.09) |
| | nCy.GAN [23] | 107.76 (11.2) | 0.60 (0.08) | 96.14 (9.3) | 0.68 (0.05) | 109.32 (10.4) | 0.68 (0.06) | 88.36 (8.2) | 0.77 (0.09) |
| | UGAC (ours) | **80.19** (10.4) | **0.78** (0.09) | **72.32** (8.4) | **0.82** (0.07) | **95.37** (9.3) | **0.77** (0.04) | **68.38** (9.8) | **0.87** (0.11) |
| Uniform | gcGAN [22] | 96.76 (18.2) | 0.66 (0.03) | 104.83 (11.7) | 0.49 (0.09) | 129.54 (15.1) | 0.47 (0.09) | 91.45 (13.3) | 0.71 (0.08) |
| | CUT [11] | 98.45 (9.8) | 0.59 (0.09) | 108.21 (7.5) | 0.53 (0.14) | 114.45 (21.9) | 0.55 (0.12) | 75.31 (8.3) | 0.78 (0.15) |
| | Cy.GAN [10] | 111.17 (15.4) | 0.35 (0.08) | 91.47 (10.8) | 0.70 (0.10) | 158.57 (25.2) | 0.39 (0.16) | 85.24 (9.5) | 0.72 (0.05) |
| | nCy.GAN [23] | 97.89 (12.1) | 0.64 (0.04) | 75.97 (10.7) | 0.78 (0.16) | 106.79 (18.7) | 0.69 (0.14) | 70.89 (8.8) | 0.81 (0.09) |
| | UGAC (ours) | **63.77** (8.5) | **0.83** (0.07) | **51.24** (6.6) | **0.88** (0.11) | **92.77** (13.2) | **0.78** (0.07) | **43.54** (6.2) | **0.89** (0.05) |
| Impulse | gcGAN [22] | 105.64 (17.3) | 0.60 (0.07) | 116.55 (15.8) | 0.45 (0.11) | 134.56 (10.7) | 0.40 (0.11) | 121.31 (17.4) | 0.66 (0.13) |
| | CUT [11] | 90.56 (11.6) | 0.52 (0.11) | 97.21 (7.8) | 0.65 (0.09) | 118.89 (15.9) | 0.52 (0.11) | 98.66 (9.7) | 0.69 (0.09) |
| | Cy.GAN [10] | 122.48 (19.6) | 0.30 (0.12) | 112.38 (9.8) | 0.62 (0.13) | 174.65 (19.2) | 0.33 (0.14) | 106.16 (14.8) | 0.67 (0.12) |
| | nCy.GAN [23] | 95.78 (10.6) | 0.61 (0.05) | 90.17 (13.2) | 0.77 (0.08) | 119.89 (12.8) | 0.57 (0.09) | 96.91 (10.57) | 0.73 (0.06) |
| | UGAC (ours) | **78.85** (6.9) | **0.80** (0.10) | **66.58** (10.4) | **0.86** (0.05) | **103.83** (9.4) | **0.72** (0.09) | **70.54** (10.4) | **0.85** (0.07) |

Table 1: Evaluating methods on four datasets under Gaussian, Uniform and Impulse perturbations, evaluated with `AMSE` (lower better) and `ASSIM` (higher better) across varying noise levels. "P" = perturbation. We show results with best performing four methods (other three are in supplementary).

**Metrics for model robustness.** We define two metrics similar to [23] to test model robustness towards noisy inputs. (i) `AMSE` is the area under the curve measuring the MSE between the outputs of the noisy input and the clean input under different levels of noise, i.e., $\texttt{AMSE} = \int_{\eta_{\min}}^{\eta_{\max}}(\texttt{MSE}(\mathcal{G}_A(a_i + \eta), \mathcal{G}_A(a_i)))d\eta$, where $\eta$ is the noise level, $\mathcal{G}_A$ denotes the generator that maps domain sample $a_i$ (from domain $A$) to domain $B$. (ii) `ASSIM` is the area under the curve measuring the SSIM [57] between the outputs of the noisy input and the clean input under different levels of noise, i.e., $\texttt{ASSIM} = \int_{\eta_{\min}}^{\eta_{\max}}(\texttt{SSIM}(\mathcal{G}_A(a_i + \eta), \mathcal{G}_A(a_i))d\eta$. These two metrics show how much the output deviates when fed with the corrupted input from the output corresponding to clean input, averaged over multiple corruption/noise levels. Computational details are in supplementary.

**Implementation details.** In our framework, the generator is a cascaded U-Net that progressively improves the intermediate features to yield high-quality output [30], we use a patch discriminator [4]. All the networks were trained using Adam optimizer [58] with a mini-batch size of 2. The initial learning rate was set to $2e^{-4}$ and cosine annealing was used to decay the learning rate over 1000 epochs. The hyper-parameters, $(\lambda_1, \lambda_2)$ (Eq. (12)) were set to $(10, 2)$. For numerical stability, the proposed network produces $\frac{1}{\alpha}$ instead of $\alpha$. The positivity constraint on the output (for predicted $\alpha, \beta$) is enforced by applying the ReLU at the end of the output layers in the network. The architecture details and the training scheme are in supplementary.

## 4.2 Comparing with the State of the Art

**Compared methods.** We compare our model to seven state-of-the-art methods for unpaired image-to-image translation, including (1) distanceGAN [59] (disGAN): a uni-directional method to map different domains by maintaining a distance metric between samples of the domains. (2) geometry consistent GAN [22] (gcGAN): a uni-directional method that imposes pairwise distance and geometric constraints. (3) UNIT [1]: a bi-directional method that matches the latent representations of the two domain. (4) CUT [11]: a uni-directional method that uses contrastive learning to match the patches in

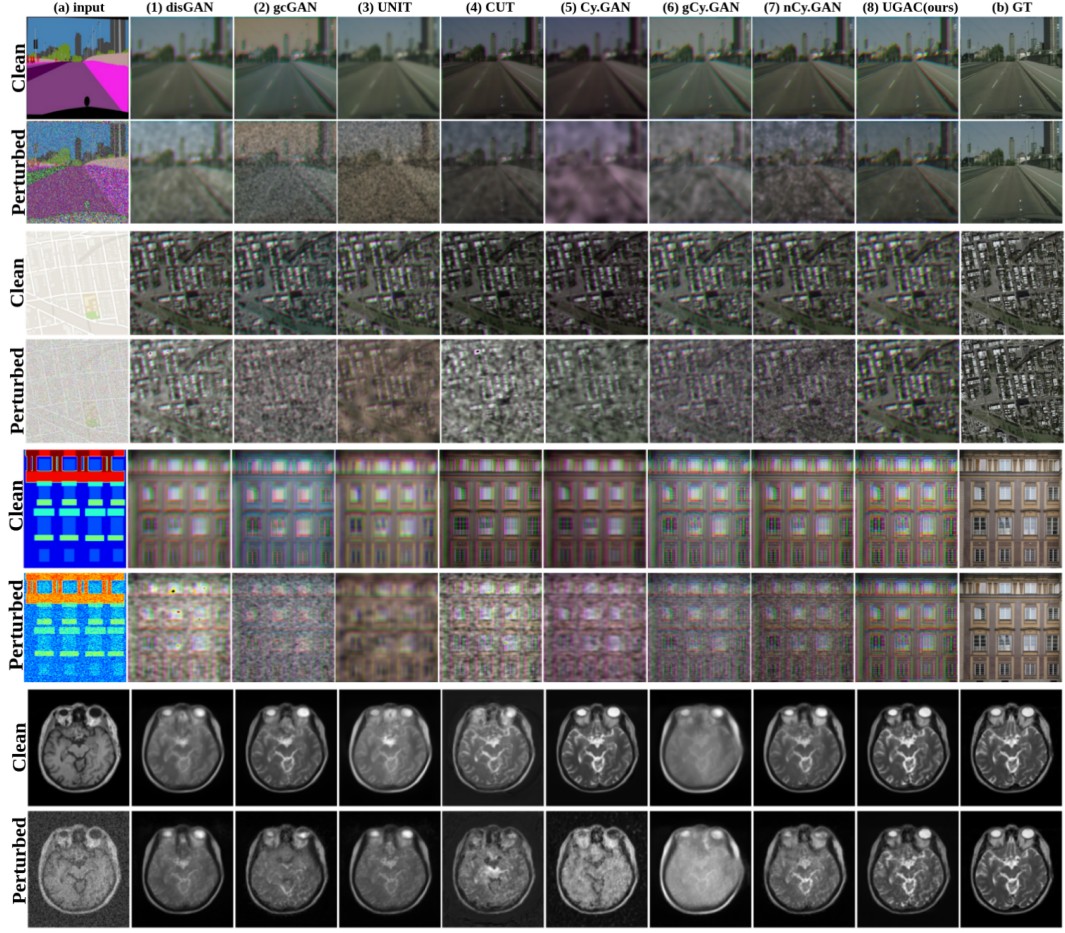

Figure 4: Qualitative results on Cityscapes, Google Maps, CMP Facade, and IXI. Outputs of clean image (at NL0) and perturbed image (at NL3) are shown. (a) input, (1)–(7) outputs from compared methods, and (8) output from UGAC, (b) ground-truth images. Outputs of UGAC are much closer to groundtruth images (better in quality) than the other methods in the presence of noise perturbations.

the same locations in both domains. (5) CycleGAN [10] (Cy.GAN): a bi-directional method that uses cycle consistency loss. (6) guess CycleGAN [23]: a variant of CycleGAN that uses an additional guess discriminator that "guesses" at random which of the image is fake in the collection of input and reconstruction images. (7) adversarial noise CycleGAN [23] (nCy.GAN): another variant of CycleGAN that introduces noise in the cycle consistency loss. Note that both guess CycleGAN [23] and adversarial noise CycleGAN [23] improve the model robustness to noise.

**Quantitative evaluation.** As described in Section 4.1, we trained the models using the *clean* images (NL0) and evaluated them at varying noise levels (NL0, NL1, NL2, NL3), results are detailed next.

Figure 3 shows the quantitative results on *Cityscapes* dataset with Gaussian perturbation. When increasing the noise levels, we observe that the performance of compared methods degrade significantly, while our method remains more robust to noise – e.g., the mean `IoU.SEGM` values are changed from around 0.24 to 0.2 for our model but degrades from around 0.24 to 0.05 for the baseline Cy.GAN. Similarly, our model outperforms two strong competitors (gCy.GAN, nCy.GAN) that are built to defend noise perturbation on higher noise levels. Similar trends are observed for other datasets (in supplementary). This indicates that our model offers better translation quality at higher noise levels.

To evaluate model robustness, we tested different methods using the metrics `AMSE` and `ASSIM` to quantify the overall image quality under increasing noise levels as defined in Section 4.1. Table 1 shows the performance of all the models on different datasets for three types of perturbations, i.e., Gaussian, Uniform, and Impulse. We can see that the proposed UGAC model performs better

than other methods. For instance, when adding Gaussian noise, UGAC obtains a much better `ASSIM` of 0.78/0.82/0.77/0.87 vs. 0.60/0.68/0.68/0.77 yielded by the best competitor nCy.GAN on Cityscapes/Maps/Facade/IXI. When adding Uniform noise or Impulse noise, we can also find that our model outperforms the other methods by substantial margins. Overall, the better performance of UGAC on different datasets suggests its stronger robustness towards various types of perturbations.

**Qualitative results.** Figure 4 visualizes the generated output images for Cityscapes, Google Maps, CMP Facade, and IXI datasets where all the models are trained with clean images and tested with either clean images or perturbed images. The test-time perturbation is of type Gaussian and corresponds to noise-level NL2. We see that, while all the methods generate samples of high quality when tested on unperturbed clean input; whereas when tested with perturbed inputs, we observe results with artifacts but the artifacts are imperceptible in our UCAC method.

The results on Cityscapes dataset (with the primary direction, translating from segmentation maps to real photo) demonstrate that with perturbed input, methods such as disGAN, gcGAN, UNIT generate images with high frequency artifacts (col.1 to 3), whereas methods such as CUT, Cy.GAN, gCy.GAN and nCy.GAN (col.4 to 7) generate images with low frequency artefacts. Both kinds of artefact lead to degradation of the visual quality of the output. Our method (col.8) generates output images that still preserve all the high frequency details and are visually appealing, even with perturbed input. Similar trends are observed for other datasets including Maps (with primary translation from maps to photo) and Facade (with primary translation from segmentation maps to real photo).

For the IXI dataset (with primary translation from T1 to T2 MRI scans), we observe that the other models fail to reconstruct medically relevant structures like trigeminal-nerve (in the centre) present in the input T1 MRI scans. Moreover, high-frequency details throughout the white and grey matter in the brain are missing. In contrast, our method gracefully reconstructs many of the high-frequency details. More qualitative results are in supplementary, with similar trends as in Figure 4. It shows that our model is capable of generating images of good quality at higher noise levels.

## 4.3 Analyzing the Model Uncertainty

**Evaluating the generalized adaptive norm.** We study the performance of our method by modelling the per-pixel residuals in three ways on IXI dataset. First, i.i.d Gaussian distribution, i.e., $(\alpha_{ij}, \beta_{ij})$ is manually set to $(1, 2) \forall i, j$, which is equivalent to using fixed $l_2$ norm at every pixel in cycle consistency loss ($\mathcal{L}_{\alpha\beta}|_{\alpha=1, \beta=2}$). visual quality when given perturbed input. Second, i.i.d Laplace distribution, i.e., $(\alpha_{ij}, \beta_{ij})$ is manually set to $(1, 1) \forall i, j$, which is equivalent to using fixed $l_1$ norm at every pixel in cycle consistency loss ($\mathcal{L}_{\alpha\beta}|_{\alpha=1, \beta=1}$). Third, independent but non-identically distributed generalized Gaussian distribution (UGAC), which is equivalent to using spatially varying $l_q$ quasi-norms where $q$ is predicted by the network for every pixel ($\mathcal{L}_{\alpha\beta}|_{\text{pred}}$).

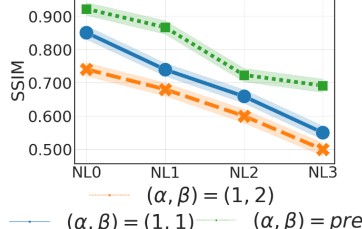

Figure 5: Adaptive $(\alpha, \beta)$=pred vs. fixed $(\alpha, \beta)$=$(1, 1)$ and $(\alpha, \beta)$=$(1, 2)$ norm.

Fig 5 shows the quantitative performance of these three variants across different noise levels for IXI datasets. We see that spatially adaptive quasi-norms perform better than fixed norms, even at higher noise levels (i.e., presence of outliers). Note that our GGD based heteroscedastic model subsumes the Gaussian ($\alpha = 1, \beta = 2$) and Laplacian ($\alpha = 1, \beta = 1$). Moreover, the heteroscedastic versions of Gaussian and Laplacian can be obtained by fixing $\beta$, i.e., for Laplacian ($\beta = 1$) and for Gaussian ($\beta = 2$), and varying $\alpha$. Modeling residuals as GGD is more liberal than both homo/hetero-scedastic Gaussian/Laplacian distribution because it is able to capture all the heavier/lighter-tailed distributions (along with all possible Gaussian/Laplacian distributions) that are beyond the modeling capabilities of Gaussian/Laplacian alone.

**Visualizing uncertainty maps.** We visualize our uncertainty maps for the T1w MRI (domain $A$) to T2w MRI (domain $B$) translation task, on IXI dataset, with perturbations in the input (NL3).

Figure 6-(a) shows input axial slices (T1w at NL3). The perturbations have degraded the high-frequency features (see green ROI). Figure 6-(b) shows the corresponding ground-truth axial slice (T2w MRI). Figure 6-(c) shows that our method recovers high-frequency details. However, we

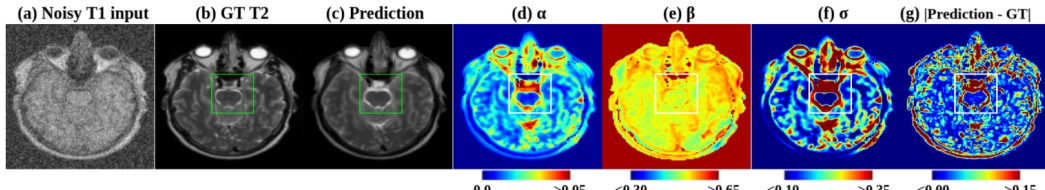

Figure 6: Visualization of uncertainty maps for noisy input at NL3 (sample from IXI test-set). **(a)** Noisy T1w MRI as input. **(b)** Corresponding ground-truth T2w MRI. **(c)** Predicted T2w MRI. **(d)-(e)** Predicted $\alpha$ and $\beta$ maps. **(f)** Uncertainty maps derived from predicted $\alpha$ and $\beta$ maps. **(g)** Absolute residual between the prediction and the ground-truth.

observe a higher contrast (compared to ground-truth) (green ROI). The subtle disparity between the contrast has been picked up by our scale-map ($\alpha$) and shape-map ($\beta$) as shown in Figure 6-(d) and (e), respectively. Moreover, we see that, although our formulation assumes independent (but non-identically) likelihood model for the pixel level residuals, the structure in the $\alpha$ and the $\beta$ (Figure 6-(d) and (e)) shows that the model learns to exploit the correlation in the neighbourhood pixels. The pixel-level variation in the $\alpha$ and $\beta$ yields pixel-level uncertainty values in the predictions as described in Section 4.1.

Figure 6-(f) shows the uncertainty map ($\sigma$) for the predictions made by the network. We see that the disparity in the contrasts between the prediction and the ground-truth is reflected as high uncertainty in the disparity region, i.e., uncertainty is high where the reconstruction is of inferior quality, indicated by high-residual values shown in Figure 6-(g). The correspondence between uncertainty maps (Figure 6-(f)) and residual maps (Figure 6-(g)) suggests that uncertainty maps can be used as the proxy to residual maps (that are unavailable at the test time, as the ground-truth images will not be available) and can serve as an indicator of image quality. More samples showing the translation results with high uncertainty ($\sigma$ estimated using $\alpha$, $\beta$) are shown in supplementary.

**Residual scores vs. uncertainty scores.** To further study the relationship between the uncertainty maps and the residual maps across a wide variety of images, we analyze the results on IXI test set. We show the density and the scatter-plot between the residual score and uncertainty score in Figure 7, where every point represents a single image. For an image, the mean residual score (on the $y$-axis) is derived as the mean of absolute residual values for all the pixels in the image. Similarly, the uncertainty score (on the $x$-axis) is calculated as the mean of uncertainty values of all the pixels in that image. From the plot, we see that across the test-set mean uncertainty score correlates positively with the mean residual score, i.e., higher uncertainty score corresponds to higher residual. An image with higher residual score represents a poor quality image. This further supports the idea that uncertainty maps derived from our method can be used as a proxy to residual that indicates the overall image quality of the output generated by our network.

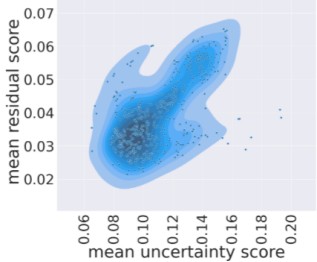

Figure 7: Residual scores vs. uncertainty scores.

## 5 Discussion and Conclusion

In this work, we propose an uncertainty-aware generalized cycle consistency for unpaired image translation along with uncertainty estimation. Our formulation assumes the pixel-wise independent (but non-identically) distributed likelihood model for the residuals, relaxing the i.i.d. assumption made by the previous work. However, our experiments also show that the model learns the underlying structure between the neighbourhood pixels and predicts the structured/correlated parameters for the output distribution (i.e., $\alpha$, $\beta$ for MRI translation shown in Figure 6-(d) and (e)).

We demonstrate the efficacy of the proposed method on robust unpaired image translation on various datasets spanning autonomous driving, maps, facades, and medical images consisting of MRI scans. We also demonstrate the robustness of our method by evaluating the performance in different kinds of perturbations in the input with varying severity and show that our method outperforms all the baselines by generating superior images in terms of quantitative metrics and appearance. In addition,

we show that the uncertainty estimates (derived from the estimated parameters of the distribution, $\alpha$ and $\beta$) are faithful proxy to the residuals between the predictions and the ground truth.

It is worth noting that robustness towards various kinds of perturbation can also be achieved by data augmentation techniques that include the perturbed images in the training phase. However, this is orthogonal to the concept proposed in this work that achieves robustness via the a new modeling technique. In principle, one could combine both the augmentation techniques and modeling techniques to obtain more robust models. In this work, we used relatively small neural networks (in terms of parameters based on UNet), while this network has not been used previously for this problem, we employ it to train our models with limited compute with reasonable training time and a lower memory footprint (details of the networks available in the Appendix A.5). This however affects the performance of the networks, and leads to images with artifacts/distortions (specially with small datasets consisting few hundred samples). Our method can be applied to deeper neural networks with more parameters/higher capacity and trained with higher resolution images, which would lead to significantly better performance, given enough compute.

An interesting avenue for further exploration is the analysis of uncertainty maps when presented with anomalous inputs, beyond perturbations, with stronger shifts between training and test data distribution which will be investigated in future.

## Broader Impact

Modern deep-learning-based image translation schemes are becoming more popular. They allow the generation of synthetic datasets, e.g., for the segmentation use case in autonomous driving, faster image acquisition via algorithmic super-resolution, image enhancement in computational photography, faster and cheaper medical diagnosis by translating between different imaging modalities. However, critical areas like medical imaging and autonomous driving require methods that are robust towards various perturbations and, at the same time, can also provide uncertainty estimates in the predictions. Estimating the uncertainty in the prediction can help trigger expert intervention preventing fatal scenarios.

We introduced a novel image-to-image translation model capable of estimating uncertainty along with the predictions and is shown to be beneficial in ensuring good image translation quality and good model performance on downstream tasks even in the presence of unseen noisy patterns in input images at inference time. Furthermore, our method is potentially applicable to detect ambiguities in the images. These merits could bring positive societal impacts to various critical application domains, such as medical imaging and autonomous driving.

**Acknowledgements** This work has been partially funded by the ERC (853489 - DEXIM) and by the DFG (2064/1 – Project number 390727645). The authors thank the International Max Planck Research School for Intelligent Systems (IMPRS-IS) for supporting Uddeshya Upadhyay.

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
