# Robustness via Uncertainty-aware Cycle Consistency

## A  Supplementary Material

We first provide the details of various kinds of perturbations followed by the method to generate different noise-levels for each of the perturbation. We then provide the computational details of the metrics used followed by code snippets for key components of our framework written in *python* using *PyTorch* deep learning library. We also present the values for various configurations and the hyper-parameter values used to train our network. We finally provide more quantitative results and output samples for various models, indicating a general trend in output for various baselines and our method as discussed in the main paper in Section-4.2.

### A.1  Perturbations

**Gaussian perturbation.** Let the image from domain $\mathcal{X}$ be represented by $x^{C \times H \times W} \in \mathcal{X}$. Gaussian perturbation samples i.i.d random variables for every pixel from a Gaussian distribution, i.e.,

$$x_G^{\text{pert}} = x + \eta_G \text{ where } \eta_G^{C \times H \times W} \sim \mathcal{N}(\mathbf{0}^{C \times H \times W}, \sigma^{C \times H \times W}). \tag{1}$$

Where $\mathcal{N}(\mathbf{0}^{C \times H \times W}, \sigma^{C \times H \times W})$ represents a multivariate Gaussian distribution with $(\mathbf{0}^{C \times H \times W}, \sigma^{C \times H \times W})$ as mean and standard deviation.

**Uniform perturbation.** Similar to above for an image $x^{C \times H \times W} \in \mathcal{X}$, Uniform perturbation samples i.i.d random variables for every pixel from a Uniform distribution, i.e.,

$$x_U^{\text{pert}} = x + \eta_U \text{ where } \eta_U^{C \times H \times W} \sim \mathcal{U}(\mathbf{0}^{C \times H \times W}, \kappa^{C \times H \times W}). \tag{2}$$

Where $\mathcal{U}(\mathbf{0}^{C \times H \times W}, \kappa^{C \times H \times W})$ represents a multivariate Uniform distribution with $(\mathbf{0}^{C \times H \times W}, \kappa^{C \times H \times W})$ as minimum/maximum values indicating the range of distribution.

**Impulse perturbation.** Similar to above for an image $x^{C \times H \times W} \in \mathcal{X}$, each pixel is, with probability $p$, replaced by an uniformly sampled random color. We achieve this computationally is by first sampling a random mask $\mathcal{M}^{1 \times H \times W} \sim \mathcal{B}(p)^{1 \times H \times W}$ where $\mathcal{B}(p)$ represents the Bernoulli distribution and a random variable sampled from $\mathcal{B}(p)$ is 1 with probability $p$ and 0 with probability $1 - p$. Then we sample a random coloured image $\mathcal{Q}^{C \times H \times W} \sim \mathcal{U}(\mathbf{0}^{C \times H \times W}, \mathbf{1}^{C \times H \times W})$. The perturbed image,

$$x_I^{\text{pert}} = \mathcal{M} \odot x + (1 - \mathcal{M}) \odot \mathcal{Q}. \tag{3}$$

### A.2  Increasing perturbation levels

**Gaussian perturbation levels.** For increasing Gaussian perturbation, we increase the magnitude of the standard deviation $\sigma$. In our experiments, the images were normalized to have pixel values between 0 to 1 and NL0, NL1, NL2 and NL3 corresponds to $\sigma = 0, \sigma = 0.10, \sigma = 0.20, \sigma = 0.30$.

**Uniform perturbation levels.** For increasing Uniform perturbation, we increase the magnitude of the maximum value $\kappa$. In our experiments, the images were normalized to have pixel values between 0 to 1 and NL0, NL1, NL2 and NL3 corresponds to $\kappa = 0, \sigma = 0.20, \sigma = 0.40, \sigma = 0.60$

**Impulse perturbation levels.** For increasing Impulse perturbation, we increase the probability of replacing a pixel with random colour ($p$). Similar to above, images were normalized bettwen 0 to 1 and NL0, NL1, NL2 and NL3 corresponds to $p = 0, p = 0.15, p = 0.30, p = 0.45$

## A.3 Computing AMSE and ASSIM

We define two metrics similar to [23] to test model robustness towards noisy inputs. (i) AMSE is the area under the curve measuring the MSE between the outputs of the noisy input and the clean input under different levels of noise, i.e.,

$$\texttt{AMSE} = \int_{\eta_{\min}}^{\eta_{\max}} \texttt{MSE}(\mathcal{G}_A(a_i + \eta), \mathcal{G}_A(a_i))d\eta, \tag{4}$$

where $\eta$ is the noise level, $\mathcal{G}_A$ denotes the generator that maps domain sample $a_i$ (from domain $A$) to domain $B$. (ii) ASSIM is the area under the curve measuring the SSIM [51] between the outputs of the noisy input and the clean input under different levels of noise, i.e.,

$$\texttt{ASSIM} = \int_{\eta_{\min}}^{\eta_{\max}} \texttt{SSIM}(\mathcal{G}_A(a_i + \eta), \mathcal{G}_A(a_i))d\eta. \tag{5}$$

To estimate both the integrals, we use the following approximation: Let there be $N$ number of noise-levels (NL0, NL1, NL2 ... NLN) corresponding to $\eta_{\min}, \eta_1, \eta_2...\eta_{N-1}, \eta_{\max}$, then the integral for AMSE is computed as,

$$\begin{aligned}
\texttt{AMSE} &= \int_{\eta_{\min}}^{\eta_{\max}} \texttt{MSE}(\mathcal{G}_A(a_i + \eta), \mathcal{G}_A(a_i))d\eta \\
&= (\eta_{\max} - \eta_{\min})\frac{1}{N}\sum_k \texttt{MSE}(\mathcal{G}_A(a_i + \eta_k), \mathcal{G}_A(a_i)).
\end{aligned} \tag{6}$$

Similarly, the integral for ASSIM is computed as,

$$\begin{aligned}
\texttt{ASSIM} &= \int_{\eta_{\min}}^{\eta_{\max}} \texttt{SSIM}(\mathcal{G}_A(a_i + \eta), \mathcal{G}_A(a_i))d\eta \\
&= (\eta_{\max} - \eta_{\min})\frac{1}{N}\sum_k \texttt{SSIM}(\mathcal{G}_A(a_i + \eta_k), \mathcal{G}_A(a_i)).
\end{aligned} \tag{7}$$

## A.4 Preprocessing of the datasets

All four datasets (Cityscapes, Maps, Facade, and IXI) are freely available to be used for research purposes and are resized so the image dimension is $256 \times 256$. All the images were also normalized so the pixel value is between 0 to 1. During the training stage we also apply random horizontal and the vertical flip transformation for data augmentation. We used Nvidia-2080ti graphics card for training.

## A.5 Implementation details

In our experiments, we used the implementation of CycleGAN provided at https://github.com/junyanz/pytorch-CycleGAN-and-pix2pix and Adversarial self defence cycleGAN provided at https://github.com/dbash/pix2pix_cyclegan_guess_noise, with Cascaded-UNet based generator similar to our method. We provide the code snippets for various components of our framework (written in python using PyTorch library). For the proposed model, we used generator that has lower number of parameters and modified to produce the parameters of the GGD.

Note that the choice of optimal hyper-parameter is affected by experiment configurations such as the resolution of the images, the dataset (color vs. black and white, number of samples, texture) as well as the architecture of the generator used. Since we train methods with UNet based generators, and on rescaled images of size 256x256 and on multiple datasets, we need to tune the hyperparameters. We follow a simple strategy to tune them, that is, we start off with the hyperparameters that are originally presented and search in its neighborhood (i.e., for loop going over discrete values in the neighborhood) for a set of parameters that lead to higher performance on the validation set.

**Residual Convolutional Block (ResConv).** This is the fundamental building block of our network. It stacks multiple convolutional layers together and exploits the benefits of residual skip connections by connecting the block's input to the output (via a convolutional layer) with an additive skip connection. Algorithm 1 shows the code snippet for the same.

**Algorithm 1** PyTorch code for Residual Conv block

```python
class ResConv(nn.Module):
    """
    Residual convolutional block (Resconv), where
    convolutional block consists: (convolution => [BN] => ReLU) * 3
    residual connection adds the input to the output
    """
    def __init__(self, in_channels, out_channels, mid_channels=None):
        super().__init__()
        if not mid_channels:
            mid_channels = out_channels
        self.res_conv = nn.Sequential(
            nn.Conv2d(in_channels, mid_channels, kernel_size=3, padding=1),
            nn.BatchNorm2d(mid_channels),
            nn.ReLU(inplace=True),
            nn.Conv2d(mid_channels, mid_channels, kernel_size=3, padding=1),
            nn.BatchNorm2d(mid_channels),
            nn.ReLU(inplace=True),
            nn.Conv2d(mid_channels, out_channels, kernel_size=3, padding=1),
            nn.BatchNorm2d(out_channels),
            nn.ReLU(inplace=True),
            nn.Dropout2d(0.05)
        )
        self.in_conv = nn.Sequential(
            nn.Conv2d(in_channels, out_channels, kernel_size=3, padding=1),
            nn.BatchNorm2d(out_channels),
            nn.ReLU(inplace=True),
        )
    def forward(self, x):
        x_in = self.in_conv(x)
        x1 = self.res_conv(x)
        return self.res_conv(x) + x_in
```

**Down-sampling Block (Down).** This block uses max-pooling operations on the input features to spatially downsample the feature maps. It is followed by a *ResConv* operation, which may change the number of channels in the output feature maps. In our case, we designed this block to down-sample the feature maps by the size of 2 across height and width. Algorithm 2 shows the code snippet.

**Algorithm 2** PyTorch code for Downsampling block

```python
class Down(nn.Module):
    """Downscaling with maxpool then Resconv"""
    def __init__(self, in_channels, out_channels):
        super().__init__()
        self.maxpool_conv = nn.Sequential(
            nn.MaxPool2d(2),
            ResConv(in_channels, out_channels)
        )
    def forward(self, x):
        return self.maxpool_conv(x)
```

**Up-sampling Block (Up).** This block performs the up-sampling operation on the input feature-maps, followed by feature concatenation with a previous feature-map passed as a second input to the module. The up-sampling procedure can be performed using interpolation or transposed-convolutional block. For our network, we use the *bilinear* interpolation performing 2x up-sampling, shown in Algorithm 3.

**Algorithm 3** PyTorch code for Upsampling block

```python
class Up(nn.Module):
    """Upscaling then Resconv"""
    def __init__(self, in_channels, out_channels, bilinear=True):
        super().__init__()
        # if bilinear, use the normal convolutions to reduce the number of channels
        if bilinear:
            self.up = nn.Upsample(scale_factor=2, mode='bilinear', align_corners=True)
            self.conv = ResConv(in_channels, out_channels, in_channels // 2)
        else:
            self.up = nn.ConvTranspose2d(in_channels , in_channels // 2, kernel_size=2, stride=2)
            self.conv = ResConv(in_channels, out_channels)
    def forward(self, x1, x2):
        x1 = self.up(x1)
        # input is CHW
        diffY = x2.size()[2] - x1.size()[2]
        diffX = x2.size()[3] - x1.size()[3]
        x1 = F.pad(
            x1,
            [
                diffX // 2, diffX - diffX // 2,
                diffY // 2, diffY - diffY // 2
            ]
        )
        x = torch.cat([x2, x1], dim=1)
        return self.conv(x)
```

**Final Convolutional Block (OutConv).** This is a convolutional block that acts as a post-processing cap at the end of the network and produces the final output with a desirable number of channels. Algorithm 4 shows the block.

**Algorithm 4** PyTorch code for final convolutional block

```python
class OutConv(nn.Module):
    """Output convolutional layer"""
    def __init__(self, in_channels, out_channels):
        super(OutConv, self).__init__()
        self.conv = nn.Conv2d(in_channels, out_channels, kernel_size=1)
    def forward(self, x):
        return self.conv(x)
```

**Simple U-Net (UNet).** This is a U-Net that utilizes the above building components to build the joint encoder and decoder part of the U-Net. Algorithm 5 shows the network.

**Algorithm 5** PyTorch code for Simple U-Net

```python
class UNet(nn.Module):
    """Simple U-net using Resconv"""
    def __init__(self, n_channels, out_channels, bilinear=True):
        super(UNet, self).__init__()
        self.n_channels = n_channels
        self.out_channels = out_channels
        self.bilinear = bilinear
        ####
        self.inc = ResConv(n_channels, 64)
        self.down1 = Down(64, 128)
        self.down2 = Down(128, 256)
        self.down3 = Down(256, 512)
        factor = 2 if bilinear else 1
        self.down4 = Down(512, 1024 // factor)
        self.up1 = Up(1024, 512 // factor, bilinear)
        self.up2 = Up(512, 256 // factor, bilinear)
        self.up3 = Up(256, 128 // factor, bilinear)
        self.up4 = Up(128, 64, bilinear)
        self.outc = OutConv(64, out_channels)
    def forward(self, x):
        x1 = self.inc(x)
        x2 = self.down1(x1)
        x3 = self.down2(x2)
        x4 = self.down3(x3)
        x5 = self.down4(x4)
        x = self.up1(x5, x4)
        x = self.up2(x, x3)
        x = self.up3(x, x2)
        x = self.up4(x, x1)
        y = self.outc(x)
        return y
```

**Cascaded U-Net (CasUNet).** Cascaded U-Net concatenates different U-Nets into a single chain. Each U-Net refines the feature maps and produces an output, slightly better than the previous U-Net. In our study, we use Cascaded U-Net with 2 component U-Nets. Algorithm 6 shows the network.

**Algorithm 6** PyTorch code for Cascaded U-Net

```python
class CasUNet(nn.Module):
    """Cascaded Simple U-net"""
    def __init__(self, n_unet, io_channels, bilinear=True):
        super(CasUNet, self).__init__()
        self.n_unet = n_unet
        self.io_channels = io_channels
        self.bilinear = bilinear
        ####
        self.unet_list = nn.ModuleList()
        for i in range(self.n_unet):
            self.unet_list.append(UNet(self.io_channels, self.io_channels, self.bilinear))
    def forward(self, x):
        y = x
        for i in range(self.n_unet):
            if i==0:
                y = self.unet_list[i](y)
            else:
                y = self.unet_list[i](y+x)
        return y
```

**U-Net for Generalized Gaussian (UNet_3head).** This is a U-Net where the last few blocks (head of the network) is split into 3 for predicting the parameters for the generalized Gaussian distribution $(\mu, \alpha, \beta)$. The positivity constraint of scale $(\alpha)$ parameter and the shape $(\beta)$ is imposed by having a ReLU activation function at the end of the respective heads. Also, for numerical stability during the training, we predict $\frac{1}{\alpha}$ instead of $\alpha$. Algorithm 7 shows the network.

**Algorithm 7** PyTorch code for modified U-Net to predict parameters of generalized Gaussian

```python
class UNet_3head(nn.Module):
    """
    U-net modified to predict the parameters of the
    generalized Gaussian distribution by splitting the
    head into three (to predict: mean, beta, 1/alpha)
    """
    def __init__(self, n_channels, out_channels, bilinear=True):
        super(UNet_3head, self).__init__()
        self.n_channels = n_channels
        self.out_channels = out_channels
        self.bilinear = bilinear
        ####
        self.inc = ResConv(n_channels, 64)
        self.down1 = Down(64, 128)
        self.down2 = Down(128, 256)
        self.down3 = Down(256, 512)
        factor = 2 if bilinear else 1
        self.down4 = Down(512, 1024 // factor)
        self.up1 = Up(1024, 512 // factor, bilinear)
        self.up2 = Up(512, 256 // factor, bilinear)
        self.up3 = Up(256, 128 // factor, bilinear)
        self.up4 = Up(128, 64, bilinear)
        #per pixel multiple channels may exist
        self.out_mean = OutConv(64, out_channels)
        #variance will always be a single number for a pixel
        self.out_alpha = nn.Sequential(
            OutConv(64, 128),
            OutConv(128, 1),
            nn.ReLU()
        )
        self.out_beta = nn.Sequential(
            OutConv(64, 128),
            OutConv(128, 1),
            nn.ReLU()
        )
    def forward(self, x):
        x1 = self.inc(x)
        x2 = self.down1(x1)
        x3 = self.down2(x2)
        x4 = self.down3(x3)
        x5 = self.down4(x4)
        x = self.up1(x5, x4)
        x = self.up2(x, x3)
        x = self.up3(x, x2)
        x = self.up4(x, x1)
        y_mean, y_1_over_alpha, y_beta = self.out_mean(x), \
            self.out_alpha(x), self.out_beta(x)
        return y_mean, y_1_over_alpha, y_beta
```

**Cascaded U-Net for Generalized Gaussian Distribution (GGD), (CasUNet_3head).** Cascaded U-Net for GGD concatenates different U-Net blocks and one UNet_3head into a single chain. Each U-Net refines the feature maps and produces an output slightly better than the previous U-Net. The final component is U-Net_3head that estimates the parameters of the GGD. In our study, we use Cascade with 2 components, UNet and UNet_3head. Algorithm 8 shows the network.

**Algorithm 8** PyTorch code for modified cascaded U-Net to predict parameters of generalized Gaussian

```python
class CasUNet_3head(nn.Module):
    def __init__(self, n_unet, io_channels, bilinear=True):
        super(CasUNet_3head, self).__init__()
        self.n_unet = n_unet
        self.io_channels = io_channels
        self.bilinear = bilinear
        ####
        self.unet_list = nn.ModuleList()
        for i in range(self.n_unet):
            if i != self.n_unet-1:
                self.unet_list.append(UNet(self.io_channels, self.io_channels, self.bilinear))
            else:
                self.unet_list.append(UNet_3head(self.io_channels, self.io_channels, self.bilinear))
    def forward(self, x):
        y = x
        for i in range(self.n_unet):
            if i==0:
                y = self.unet_list[i](y)
            else:
                y = self.unet_list[i](y+x)
        y_mean, y_1_over_alpha, y_beta = y[0], y[1], y[2]
        return y_mean, y_1_over_alpha, y_beta
```

**Patch Discriminator (NLayerDiscriminator).** This is the discriminator used with the generator for the adversarial training. As described in the main paper (Section-XX), instead of classifying entire image as "real" or "fake", it classifies overlapping $70 \times 70$ patches as "real" or "fake".

**Algorithm 9** PyTorch code for Patch Discriminator used for all the models

```python
class NLayerDiscriminator(nn.Module):
    """Defines a PatchGAN discriminator"""
    def __init__(self, input_nc, ndf=64, n_layers=3, norm_layer=nn.BatchNorm2d):
        """Construct a PatchGAN discriminator
        Parameters:
            input_nc (int)  -- the number of channels in input images
            ndf (int)       -- the number of filters in the last conv layer
            n_layers (int)  -- the number of conv layers in the discriminator
            norm_layer      -- normalization layer
        """
        super(NLayerDiscriminator, self).__init__()
        if type(norm_layer) == functools.partial:  # no need to use bias as BatchNorm2d has affine parameters
            use_bias = norm_layer.func == nn.InstanceNorm2d
        else:
            use_bias = norm_layer == nn.InstanceNorm2d
        kw = 4
        padw = 1
        sequence = [nn.Conv2d(input_nc, ndf, kernel_size=kw, stride=2, padding=padw), nn.LeakyReLU(0.2, True)]
        nf_mult = 1
        nf_mult_prev = 1
        for n in range(1, n_layers):  # gradually increase the number of filters
            nf_mult_prev = nf_mult
            nf_mult = min(2 ** n, 8)
            sequence += [
                nn.Conv2d(ndf * nf_mult_prev, ndf * nf_mult, kernel_size=kw, stride=2, padding=padw, bias=use_bias),
                norm_layer(ndf * nf_mult),
                nn.LeakyReLU(0.2, True)
            ]
        nf_mult_prev = nf_mult
        nf_mult = min(2 ** n_layers, 8)
        sequence += [
            nn.Conv2d(ndf * nf_mult_prev, ndf * nf_mult, kernel_size=kw, stride=1, padding=padw, bias=use_bias),
            norm_layer(ndf * nf_mult),
            nn.LeakyReLU(0.2, True)
        ]
        sequence += [nn.Conv2d(ndf * nf_mult, 1, kernel_size=kw, stride=1, padding=padw)]  # output 1 channel prediction map
        self.model = nn.Sequential(*sequence)
    def forward(self, input):
        """Standard forward."""
        return self.model(input)
```

**Training details and Hyper-parameters.** We apply the following techniques to stabilize our model training procedure. First, for GAN, we use a least-squares loss for adversarial term as explained in the main text, Equation 11 and 12. This loss is more stable during training and generates higher quality results. In particular, the adversarial loss for the generators ($\mathcal{L}^G_{\mathrm{adv}}$) is,

$$\mathcal{L}^G_{\mathrm{adv}} = \mathcal{L}_2(\mathcal{D}^A(\hat{b}_i), 1) + \mathcal{L}_2(\mathcal{D}^B(\hat{a}_i), 1). \tag{8}$$

The loss for discriminators ($\mathcal{L}^D_{\mathrm{adv}}$) is,

$$\mathcal{L}^D_{\mathrm{adv}} = \mathcal{L}_2(\mathcal{D}^A(b_i), 1) + \mathcal{L}_2(\mathcal{D}^A(\hat{b}_i), 0) + \mathcal{L}_2(\mathcal{D}^B(a_i), 1) + \mathcal{L}_2(\mathcal{D}^B(\hat{a}_i), 0). \tag{9}$$

Second, to reduce model oscillation, we update the discriminators using a history of generated images rather than those produced by the latest generators. We keep an image buffer that stores the 20 previously created images. The hyper-parameters, $(\lambda_1, \lambda_2)$ were set to $(10, 2)$. We use the Adam optimizer with $(\beta_1, \beta_2) = (0.9, 0.99)$ with a batch size of 4. All networks were trained from scratch with a learning rate of 0.0002 with cosine annealing over 1000 epochs.

### A.6 More Quantitative Results

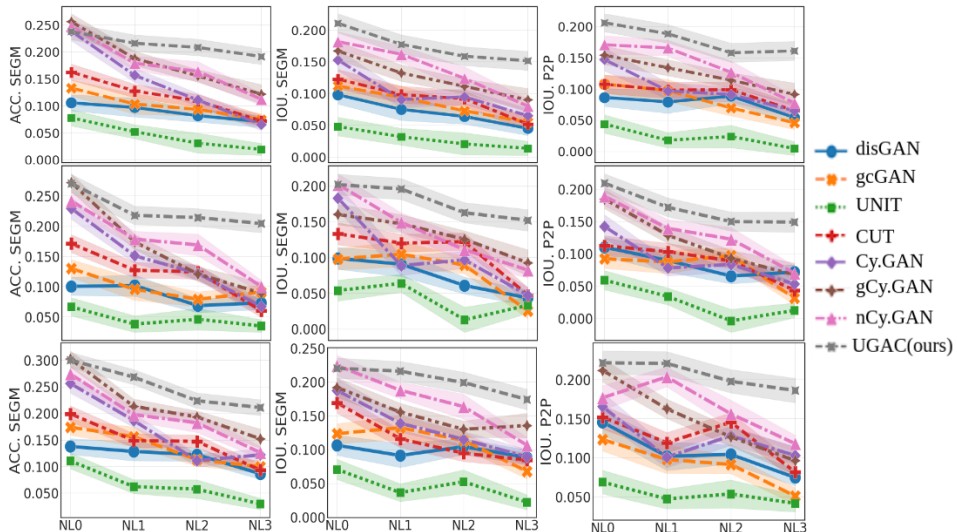

Figure 1: Evaluation of different methods on Cityscapes, Facade, and Maps with Gaussian perturbation under varying noise levels. NL0 denotes clean images without noise, NL1, NL2, NL3 are unseen noise levels. `ACC.Segm`, `IoU.Segm`, `IoU.P2P` are three metrics for evaluating translation quality. Higher is better.

We use the following baselines in the study, (1) distanceGAN (disGAN): a uni-directional method to map different domains by maintaining a distance metric between samples of the domain with the

| P | Methods | Cityscapes | | Maps | | Facade | | IXI | |
|---|---|---|---|---|---|---|---|---|---|
| | | AMSE (std)↓ | ASSIM (std)↑ | AMSE (std)↓ | ASSIM (std)↑ | AMSE (std)↓ | ASSIM (std)↑ | AMSE (std)↓ | ASSIM (std)↑ |
| Gaussian | disGAN[53] | 119.75 (13.5) | 0.49 (0.11) | 110.32 (8.6) | 0.41 (0.01) | 137.47 (12.5) | 0.46 (0.07) | 104.68 (11.8) | 0.69 (0.08) |
| | gcGAN[22] | 107.83 (10.8) | 0.62 (0.09) | 117.21 (10.6) | 0.43 (0.07) | 138.21 (11.5) | 0.41 (0.05) | 108.32 (8.7) | 0.67 (0.12) |
| | UNIT[1] | 114.78 (9.2) | 0.40 (0.08) | 109.27 (9.7) | 0.59 (0.11) | 121.45 (12.5) | 0.48 (0.12) | 114.12 (9.7) | 0.63 (0.06) |
| | CUT[11] | 108.34 (8.7) | 0.51 (0.12) | 119.32 (8.9) | 0.51 (0.11) | 123.22 (17.6) | 0.58 (0.09) | 87.12 (10.4) | 0.64 (0.07) |
| | Cy.GAN[10] | 121.32 (10.3) | 0.31 (0.13) | 107.32 (7.5) | 0.61 (0.13) | 134.23 (15.3) | 0.45 (0.07) | 98.14 (9.1) | 0.70 (0.09) |
| | gCy.GAN[23] | 113.43 (8.3) | 0.51 (0.09) | 104.42 (9.3) | 0.63 (0.08) | 124.55 (12.5) | 0.67 (0.03) | 96.12 (7.9) | 0.73 (0.07) |
| | nCy.GAN[23] | 107.76 (11.2) | 0.60 (0.08) | 96.14 (9.3) | 0.68 (0.05) | 109.32 (10.4) | 0.68 (0.06) | 88.36 (8.2) | 0.77 (0.09) |
| | UGAC (ours) | **80.19** (10.4) | **0.78** (0.09) | **72.32** (8.4) | **0.82** (0.07) | **95.37** (9.3) | **0.77** (0.04) | **68.38** (9.8) | **0.87** (0.11) |
| Uniform | disGAN[53] | 113.86 (12.1) | 0.51 (0.08) | 102.47 (9.4) | 0.43 (0.12) | 148.31 (21.3) | 0.44 (0.11) | 94.77 (16.3) | 0.73 (0.05) |
| | gcGAN[22] | 96.76 (18.2) | 0.66 (0.03) | 104.83 (11.7) | 0.49 (0.09) | 129.54 (15.1) | 0.47 (0.09) | 91.45 (13.3) | 0.71 (0.08) |
| | UNIT[1] | 100.85 (10.6) | 0.43 (0.05) | 90.21 (14.9) | 0.72 (0.10) | 139.44 (19.3) | 0.45 (0.11) | 98.31 (10.2) | 0.68 (0.10) |
| | CUT[11] | 98.45 (9.8) | 0.59 (0.09) | 108.21 (7.5) | 0.53 (0.14) | 114.45 (21.9) | 0.55 (0.12) | 75.31 (8.3) | 0.78 (0.15) |
| | Cy.GAN[10] | 111.17 (15.4) | 0.35 (0.08) | 91.47 (10.8) | 0.70 (0.10) | 158.57 (25.2) | 0.39 (0.16) | 85.24 (9.5) | 0.72 (0.05) |
| | gCy.GAN[23] | 102.52 (13.2) | 0.58 (0.03) | 92.35 (9.9) | 0.68 (0.08) | 118.55 (22.6) | 0.61 (0.09) | 81.16 (6.9) | 0.78 (0.12) |
| | nCy.GAN[23] | 97.89 (12.1) | 0.64 (0.04) | 75.97 (10.7) | 0.78 (0.16) | 106.79 (18.7) | 0.69 (0.14) | 70.89 (8.8) | 0.81 (0.09) |
| | UGAC (ours) | **63.77** (8.5) | **0.83** (0.07) | **51.24** (6.6) | **0.88** (0.11) | **92.77** (13.2) | **0.78** (0.07) | **43.54** (6.2) | **0.89** (0.05) |
| Impulse | disGAN[53] | 124.77 (10.3) | 0.46 (0.09) | 113.44 (8.6) | 0.40 (0.08) | 155.45 (14.5) | 0.41 (0.07) | 115.63 (13.7) | 0.65 (0.09) |
| | gcGAN[22] | 105.64 (17.3) | 0.60 (0.07) | 116.55 (15.8) | 0.45 (0.11) | 134.56 (10.7) | 0.40 (0.11) | 121.31 (17.4) | 0.66 (0.13) |
| | UNIT[1] | 94.87 (9.6) | 0.40 (0.04) | 110.32 (10.8) | 0.66 (0.07) | 148.32 (12.6) | 0.39 (0.13) | 133.78 (17.5) | 0.57 (0.07) |
| | CUT[11] | 90.56 (11.6) | 0.52 (0.11) | 97.21 (7.8) | 0.65 (0.09) | 118.89 (15.9) | 0.52 (0.07) | 98.66 (9.7) | 0.69 (0.09) |
| | Cy.GAN[10] | 122.48 (19.6) | 0.30 (0.12) | 112.38 (9.8) | 0.62 (0.13) | 174.65 (19.2) | 0.33 (0.14) | 106.16 (14.8) | 0.67 (0.12) |
| | gCy.GAN[23] | 125.57 (17.8) | 0.56 (0.06) | 97.46 (8.1) | 0.71 (0.06) | 149.93 (24.7) | 0.48 (0.12) | 100.94 (13.8) | 0.70 (0.07) |
| | nCy.GAN[23] | 95.78 (10.6) | 0.61 (0.05) | 90.17 (13.2) | 0.77 (0.08) | 119.89 (12.8) | 0.57 (0.09) | 96.91 (10.57) | 0.73 (0.06) |
| | UGAC (ours) | **78.85** (6.9) | **0.80** (0.10) | **66.58** (10.4) | **0.86** (0.05) | **103.83** (9.4) | **0.72** (0.09) | **70.54** (10.4) | **0.85** (0.07) |

Table 1: Evaluating methods on four datasets under Gaussian, Uniform and Impulse perturbations, evaluated with AMSE (lower better) and ASSIM (higher better) across varying noise levels. "P" = perturbation.

help of a GAN framework. (2) geometry consistent GAN (gcGAN): a uni-directional method that imposes pairwise distance and geometric constraints. (3) UNIT: a bi-directional method that matches the latent representations of the two domain. (4) CUT: a uni-directional method that uses contrastive learning to match the patches in the same locations in both domains. (5) CycleGAN (Cy.GAN): a bi-directional method that uses cycle consistency loss. (6) guess Cycle GAN: a variant of CycleGAN that uses an additional guess discriminator that "guesses" at random which of the image is fake in the collection of input and reconstruction images. (7) adversarial noise GAN (nCy.GAN): another variant of CycleGAN that introduces noise in the cycle consistency loss. To ensure a fair comparison, we use the same generator and discriminator architectures for all methods.

We trained the models using the *clean* images (NL0) and evaluated them at varying noise levels (NL0, NL1, NL2, NL3), results are detailed next.

Figure 1 shows the quantitative results on *Cityscapes* dataset with Gaussian perturbation. When increasing the noise levels, we observe that the performance of compared methods degrade significantly, while our method remains more robust to noise – e.g., the mean IoU.SEGM values are changed from around 0.24 to 0.2 for our model but degrades from around 0.24 to 0.05 for the baseline Cy.GAN. Similarly, our model outperforms two strong competitors (gCy.GAN, nCy.GAN) that are built to defend noise perturbation on higher noise levels. Similar trends are observed for other datasets

To evaluate model robustness, we tested different methods using the metrics AMSE and ASSIM to quantify the overall image quality under increasing noise levels. Table 1 shows the performance of all the models on different datasets for three types of perturbations, i.e., Gaussian, Uniform, and Impulse. We can see that the proposed UGAC model performs better than other methods. For instance, when adding Gaussian noise, UGAC obtains a much better ASSIM of 0.78/0.82/0.77/0.87 vs. 0.60/0.68/0.68/0.77 yielded by the best competitor nCy.GAN on Cityscapes/Maps/Facade/IXI. When adding Uniform noise or Impulse noise, we can also find that our model outperforms the other methods by substantial margins. Overall, the better performance of UGAC on different datasets suggests its stronger robustness towards various types of perturbations.

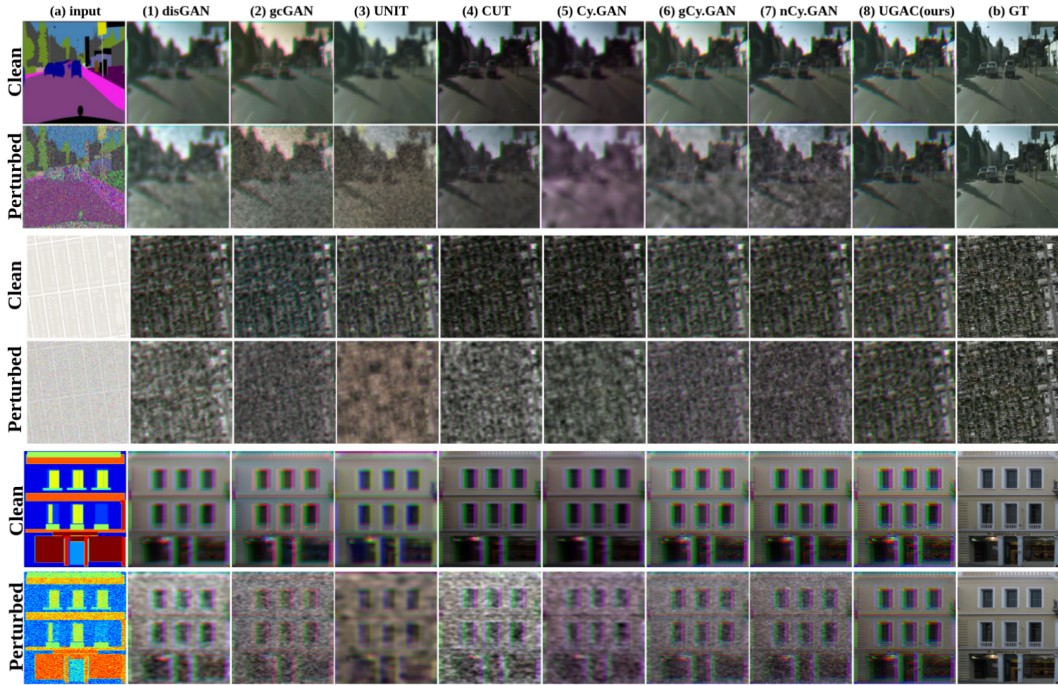

Figure 2: Qualitative results on Cityscapes, Google Maps, CMP Facade. Outputs of clean image (at NL0) and perturbed image (at NL3) are shown. **(a)** input, **(1)–(7)** outputs from compared methods, and **(8)** output from UGAC, **(b)** ground-truth images. Outputs of UGAC are much closer to groundtruth images (better in quality) than the other methods in the presence of noise perturbations.

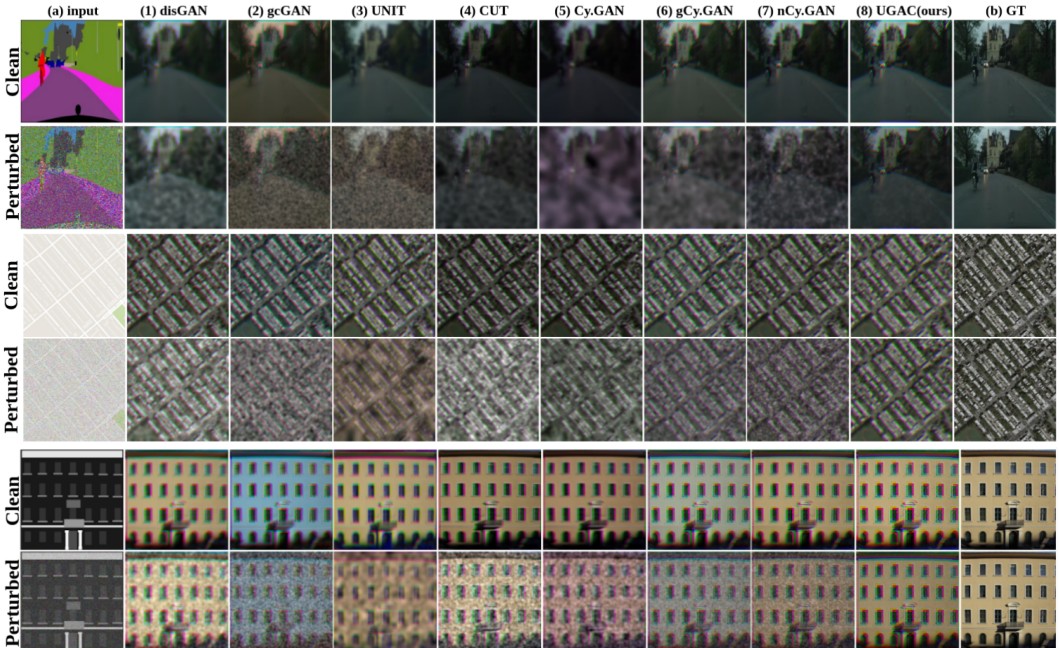

Figure 3: Qualitative results on Cityscapes, Google Maps, CMP Facade. Similar to Figure 2.

## A.7 More Qualitative Results

In this section we present more qualitative results for all the baselines that were used in the study presented in main manuscript along with our proposed model.

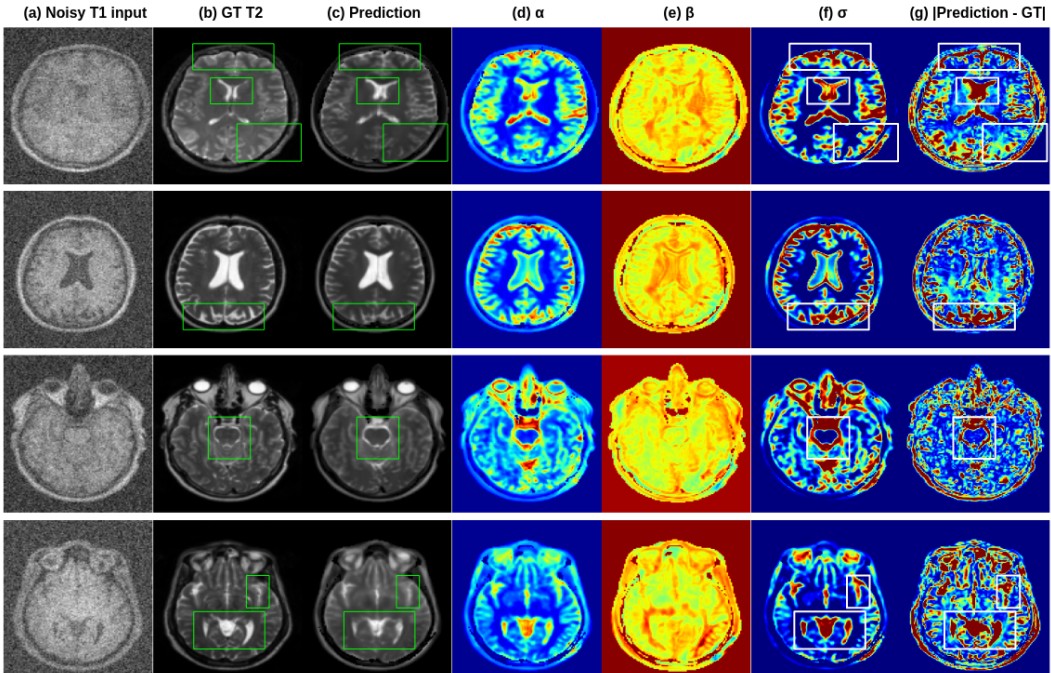

| (a) Noisy T1 input | (b) GT T2 | (c) Prediction | (d) α | (e) β | (f) σ | (g) |Prediction - GT| |

Figure 4: Samples from MRI translation with parameter, uncertainty, and residual maps

Figure 2 and 3 visualizes the generated output images for Cityscapes, Google Maps, CMP Facade where all the models are trained with clean images and tested with either clean images or perturbed images. The test-time perturbation is of type Gaussian and corresponds to noise-level NL2. We see that, while all the methods generate samples of high quality when tested on unperturbed clean input; whereas when tested with perturbed inputs, we observe results with artifacts but the artifacts are imperceptible in our UCAC method.

The results on Cityscapes dataset (with the primary direction, translating from segmentation maps to real photo) demonstrate that with perturbed input, methods such as disGAN, gcGAN, UNIT generate images with high frequency artifacts (col.1 to 3), whereas methods such as CUT, Cy.GAN, gCy.GAN and nCy.GAN (col.4 to 7) generate images with low frequency artefacts. Both kinds of artefact lead to degradation of the visual quality of the output. Our method (col.8) generates output images that still preserve all the high frequency details and are visually appealing, even with perturbed input. Similar trends are observed for other datasets including Maps (with primary translation from maps to photo) and Facade (with primary translation from segmentation maps to real photo).

Figure 4 shows the results of MRI translation going from T1 MRI to T2 MRI along with the predicted parameters, i.e., the scale ($\alpha$) and the beta ($\beta$) and also the derived uncertainty maps ($\sigma$) that are high, and the residual maps. We see that uncertainty maps are positively correlated with the residual maps that measure the error per-pixel between the predictions and the groundtruth images.