# OpenReview forum: "Robustness via Uncertainty-aware Cycle Consistency"
_NeurIPS.cc/2021/Conference — NeurIPS 2021 Poster_

### Official Review · Reviewer_TSoT · 2021-07-12

**Rating:** 6
**Confidence:** 5

**Summary:**

Authors propose a relaxation of the cycle consistency loss that takes into account per-pixel prediction uncertainty. More specifically, authors assume that cycle-reconstruction residuals are distributed as iid generalized Gaussian random variables, and maximize likelihood of these reconstruction residuals given predicted per-pixel parameters. Authors report two metrics of sensitivity to noise and three segmentation accuracy metrics across four real datasets with known GT and seven baseline methods, confirming that in all cases their method outperforms or performs on par with SotA. Authors also show that restricting parameters of their generalized Gaussian log-likelihood loss to constant values (effectively reducing it to standard CycleGAN) degrades the performance of their method according to these performance metrics. Moreover, authors showed that the overall prediction uncertainty strongly correlates with the magnitude of the true prediction error computed using GT.

**Limitations And Societal Impact:**

Authors do not explicitly discuss scenarios in which their method might fail. One possible cause for concern are over-confident predictions in the context of human computer interaction: if a machine _usually_ accurately predicts how uncertain its predictions are, and human learns to rely on there prediction uncertainty estimates, how do we mitigate the consequences of (inevitably, eventually) machine wrongly over-estimating its confidence for an unusual input, and human blindly relying on it (because "the machine said it was confident, so we decided that double-checking is not necessary")?

**Main Review:**

Writing is very clear, and the overall presentation is great. Figure 1 might need some more love - it made understanding the paper harder, not easier, but I still managed to understand everything well from text, so it is not a deal breaker.

Authors quantitatively verified all claims made in the paper and provide an exhaustive set of [baselines x datasets x metrics]. Additional meaningful ablations and visualizations (e.g. residual-vs-uncertainty plot) supporting their claims.

While I support accepting this paper, and reported results are already conclusive, I believe that following additions and clarifications would help in making these results even more convincing.

First of all, authors should directly point readers to Section A.6 of the supplementary for results of all methods on all dataset both in text and in figure captions, because my initial review assumed that these cases were not reported at all. While talking about the supplementary, “set all the CycleGAN parameters to the default ones provided in the implementation except for the weights of the cycle-consistency loss” - could you elaborate on why/how you decided to change baseline hyper-parameters from default ones?

Many images (including tested g/nCyGAN and proposed UGAC) in Figure 4 (and Figure 2 in supplementary) look like their color channels were displaced (e.g. three displaced r/g/b blobs instead of a single white blob where the car logo goes). This kind of artifact seems to be not very typical for image-to-image models and not present in any prior work. Could you please elaborate on why you think it showed up in your results (including prior work g/nCyGAN)? It is just a visualization issue or it might have affected reported results?

How does the amplitude of the noise used during training of nCyGAN relate to NL0-3 levels used during evaluation? Does changing the training noise level in nCyGAN affect the results reported in Table 1?

I understand that the generalized Gaussian is used to model heavy tales of the residuals, but does this choice matter in practise? For example, if you modeled per-pixel reconstruction residuals as iid Gaussian/Laplace rvs with predicted _per-pixel standard deviations_, did you check how that would affect the performance experimentally?

“Residual scores vs. uncertainty scores” (Figure 7) - while it is clear from this picture that there must be some significant correlation between the two, the p-value of a simple statistical test verifying that the observed deviation of the correlation coefficient from zero is significant (given your sample size) would support this point quantitatively.

Both the original cyclegan loss and the proposed loss assume that all residuals are independent, which is clearly not the case in practise. I would appreciate an extended discussion of consequences of such choice versus maximizing the likelihood of a proper conditional image density model. In other words: why modeling the residual likelihood as iid noise (gaussian or otherwise) is preferred over modeling it as a proper density model that takes pixel interactions into account.

Definitions of AMSE and ASSIM need some clarification. Is it \int |MSE(GA(ai), bi) - MSE(GA(ai + noise), bi)| or |MSE(GA(ai), GA(ai + noise))| or something else entirely?

Fig 5 - colored lines in the legend are too thin and therefore hard to see.

**Time Spent Reviewing:**

6h

---

> ### Author Response · Authors · 2021-08-09
> **Response to reviewers**
>
> We sincerely thank the reviewer for acknowledging that our paper is well-written with great presentation and recognizing our analysis as exhaustive. We will update Figure 1 and the pointer to A.6 in the appendix at appropriate places. We address the remaining comments below.
>
> ### Q: P-value for the correlation between the mean residual and mean uncertainty score to be significantly higher than 0.
> As requested we perform the following experiment to derive the P-value for the correlation coefficients. We change the seeds (0 to 1000) with NL2 Gaussian noise in the input MRI images, and calculate the correlation coefficient between the mean residual and mean uncertainty values using all the images in the test set, for each seed. This gives us a set of 1000 correlation values with a mean=0.395 and a std.dev=0.043. To establish the statistical significance, we perform one-sample t-test (i.e., statistical test to determine if the mean of an unknown population is different from a specific value). We provide results for multiple null hypotheses (HOs) in the following table.
>
> H0: population mean = $\mu$ (Null hypothesis)
>
> | $\mu$ | t-statistic | p-value | P-value < 0.05 |  Null hypothesis rejected? |
> |:---------:|:-------------:|:----------:|:--------------------:|:------------------------------------:|
> 0.000 | 286.985 | 0 | Yes | Yes |
> 0.350 | 33.346 | 1.75e-164 | Yes | Yes |
> 0.394 | 1.461 | 0.144 | No | No |
> 0.398 | -1.439 | 0.151 | No | No |
> 0.400 | -2.887 | 0.003 | Yes | Yes |
>
> From the above table, we make the following conclusions: (1) correlation value is significantly higher than 0 as the corresponding null hypothesis gets rejected, even hypothesis with mean of 0.35 gets rejected. (2) Only when the population mean is between 0.394 and 0.398, the hypothesis gets accepted. Moreover, the null hypothesis gets rejected for higher values of the population mean (> 0.398) i.e., the population mean is significantly closer to [0.394, 0.398] which shows a significant positive correlation between mean residual and mean uncertainty values.
>
> ### Q: Amplitude of noise used in training of nCy.GAN and its effect.
> Yes, the amplitude of noise used during the nCy.GAN [a] is an important parameter. If the amplitude is “too high”, the input images will be too noisy and the network will not be able to reconstruct good quality images as output. But if the amplitude of the noise is “too low” then the network will not be very robust to the high amplitude of noise during testing.
> In the original work [a], the authors introduce the Gaussian noise in the training, however we have not found a discussion on how to pick an optimal amplitude of noise during the training. We believe this is an empirical analysis that depends on the dataset and prior knowledge about the magnitude of maximum noise at the test time. However, knowing the magnitude of the maximum noise or the source/form of noise beforehand is not possible for many practical applications such as in medical imaging. Moreover, a training augmented with the Gaussian noise (like nCy.GAN) will not help in the case of other types of noise perturbations at the test time, like Uniform and Impulse perturbation, as indicated in our experiments.
>
> - [a] Adversarial Self-Defense for Cycle-Consistent GANs, Dina Bashkirova, Ben Usman, and Kate Saenko, NeurIPS19
>
> ### Q: Independence of residuals in cycleGAN and proposed loss. Discussion of conditional density models vs MLE
> Yes, it is correct that in practice residuals are not independent but typically we consider that modeling the pixel interactions using probabilistic models such as Conditional Random Fields [a][b] are generally required in the scenarios when the network needs to model the context information that takes pixel locations into account, such as image segmentation [a] and object detection [b] tasks. However, in the circumstance to cope with unseen noisy perturbation at test time, the noisy information is generally not presented with a certain structure, nor correlated with the pixel locations, so starting off with the liberal assumption of pixel-wise independence makes sense.
>
> Moreover, we would like to re-emphasize the focus to the $\alpha$ and the $\beta$ maps predicted by our method and visualized in Figure 6 (images d and e, respectively) on page 9 for the MRI dataset. We see that the values of the predicted parameters are not pixel-wise independent and show a meaningful structure. This means, even though we start off with a liberal assumption of pixel-wise independence for residuals in our method, the network learns to model the underlying pixel interactions.
>  - [a] Multiscale Conditional Random Fields for Image Labeling, CVPR2004
>  - [b] Structured Modeling of Joint Deep Feature and Prediction Refinement for Salient Object Detection, ICCCV2019
>
> ### Q: Definitions of AMSE and ASSIM need some clarification.
> Thanks for pointing this out, we would like to clarify and explain the motivation behind this metric with a concrete example.
>
> Consider an input image $a_1$ in domain A, the corresponding groundtruth image $b_1$ in domain B. Consider a generator $G_A$ that translates $a_1$ to an image in domain $B$.
> We first translate the unperturbed image (NL0) and measure two quality metrics:
> - MSE($G_A(a_1), b_1$)
> - SSIM($G_A(a_1), b_1$)
>
> These two metrics show how much the output deviates from the groundtruth image.
> Now, at higher noise levels (NL1, NL2, etc) corresponding to some perturbation $\eta$ we measure the following two metrics:
> - MSE($G_A(a_1+ \eta), G_A(a_1)$)
> - SSIM($G_A(a1+\eta), G_A(a_1)$)
>
> These two metrics show how much the output deviates when fed with the corrupted input from the output corresponding to clean input.
> We compute these metrics at every noise level (corresponding to different values), and then compute the final AMSE and ASSIM scores according to the equations :
>
> $AMSE = \int_{\eta_{min}}^{\eta_{max}} MSE(G_A(a_1+ \eta), G_A(a_1))$ and
> $ASSIM = \int_{\eta_{min}}^{\eta_{max}} SSIM(G_A(a_1+ \eta), G_A(a_1))$
>
> We will clarify this more thoroughly in the manuscript.
>
> ### Q: Practical usage of GGD and heteroscedastic Gaussian/Laplacian (i.e., predicting per-pixel variance of distribution)
> We point to Figure 5 on page 8 of our manuscript, the experiment explicitly studies the benefit of $\mathcal{L}_1$ norm over $\mathcal{L}_2$ norm and GGD based norm (with predicted parameters) over $\mathcal{L}_1$ norm. While we did not use the heteroscedastic models for Gaussian/Laplacian for comparison, we want to point out that our GGD based heteroscedastic model covers the Gaussian ($\alpha=1, \beta=2$) and Laplacian ($\alpha=1, \beta=1$) case as well. Moreover, the heteroscedastic versions of Gaussian and Laplacian can be obtained by fixing $\beta$, $\beta=1$ for Laplacian and $\beta=2$ for Gaussian, and varying $\alpha$.
>
> ### Q: Hyperparameter selections.
> Note that the choice of optimal hyper-parameter is affected by experiment configurations such as the resolution of the images, the dataset (color vs. black and white, number of samples, texture) as well as the architecture of the generator used. Since we train methods with UNet based generators, and on rescaled images of size 256x256 and on multiple datasets, we need to tune the hyperparameters. We follow a simple strategy to tune them, that is, we start off with the hyperparameters that are originally presented and search in its neighborhood (i.e., for loop going over discrete values in the neighborhood) for a set of parameters that lead to higher performance on the validation set. Libraries such as Optuna (https://optuna.org/) helps to perform such sweeps systematically and easily.
>
> ### Q: Many images look like color channels are displaced.
> The color distortions are caused by the following two reasons. (1) The color information is not encoded in the conditional input (i.e. a segmentation mask). As the segmentation mask only provides structural information, it is infeasible for the model to hallucinate the colors that are exactly in line with the groundtruth images. Moreover, the distortions in the channels happen in the high-frequency regions (like the car logo), which again is not encoded in the conditional input (2) Small scale dataset that covers a wide variety of colors and shapes make the learning of such features difficult, especially in the case where images are downsampled for training to limit the memory footprint. Please note that for the more homogenous dataset (in terms of colors/textures/shape) like Google Maps and MRI such problems do not occur.
>
> In our visualization, half of the images are generated based on noisily perturbed input which could lead to severe blurriness. When adding unseen noisy perturbation at test time, we can observe that many other existing methods are likely to degrade significantly and generate images of poor quality, i.e. severe blurriness and distortions, this happens because their learning scheme does not account for robustness.
>
> In the following table, we present the FID scores (unaffected by minor local distrotions) for the Facade dataset. We include all the methods at all the noise levels (NL0 to NL3) and the final average FID across noise levels for all the methods. The results are presented for Gaussian distortions. Note that while other methods degrade significantly and obtain much higher FID scores under different levels of noisy perturbations, UGAC obtains a more robust and better model performance under varying noisy levels.
>
> | Methods | NL0 | NL1 | NL2 | NL3 | Avg. FID |
> |:-----------:|:------:|:-----:|:------:|:----:|:------------:|
> | disGAN | 129.1 |155.7 | 189.6 | 236.1 | 177.6|
> | gcGAN | 134.7 | 178.9 | 211.8 | 257.2 | 195.7|
> | UNIT | 126.2 | 152.5 | 207.4 | 248.6 | 183.7|
> | CUT | 93.8 | 137.6 | 193.3 | 235.8 | 165.1 |
> | Cy.GAN | 100.3 | 148.2 | 210.8 | 253.7 | 178.3 |
> | gCy.GAN | 112.7 | 133.4 | 190.7 | 220.2 | 164.3 |
> | nCy.GAN | 107.5 | 128.3 | 169.6 | 194.3 | 149.9 |
> | UGAC | 95.2 | 99.6 | 107.8 | 114.7 | 104.3|

---

### Official Review · Reviewer_xn61 · 2021-07-16

**Rating:** 6
**Confidence:** 4

**Summary:**

The paper achieves improved and more robust unpaired image-to-image translation results by modeling uncertainty of the outputs. The proposed method generalized and treats the original pixel cycle loss of CycleGAN as the special case in which the uncertainty distribution is uniformly fixed. Furthermore, the predicted uncertainty can be used as a way to measure risk at inference time.

**Limitations And Societal Impact:**

I believe improving robustness and modeling uncertainty of image-to-image translation methods is certainly beneficial, especially for applications like medical imaging. The authors also adequately discuss this in Broader Impact.

**Main Review:**

The paper proposes a clear idea that improves on previous methods in terms of both output quality and robustness. The paper is well motivated and explained. Moreover, modeling uncertainty can be beneficial in assessing the risk when processing out-of-distribution, corrupted images, and the paper proves this point on medical imaging. The model uncertainty is sufficiently analyzed in various types of artifical noises via several graphs and visualizations.

For the biggest weakness, something seems wrong with the qualitative comparison results. The RGB channels seem shifted and not aligned (clean Facade results of Figure 4). Images are also quite blurry. In Fig3 of Supp Mat, the input Facade segmentation layouts are greyscale. They also look too low resolution for 256x256 images. If these images are indeed faulty, I wonder if the evaluations are also based on these images. Since there is possibility of faulty evaluation, I am hesitant to conclude a positive rating for this submission.

The evaluations for the generated images in image-to-image translation tasks often involve FID. Including FID to the paper can also make the paper more comprehensive and comparable to existing papers in this field.

The architecture of the proposed method is based on cascaded U-Net, and some changes may be coming from the architectural changes. I am curious of the results of using this architecture, but using the original cycle loss without uncertainty modeling.

Lastly, it would be interesting to see which input images of the dataset are considered as uncertain. Visualizing the top-10 most certain and uncertain examples would be interesting.

**Time Spent Reviewing:**

2.5 hours

---

> ### Author Response · Authors · 2021-08-09
> **Response to reviewer**
>
> We sincerely thank the reviewer for suggesting improvements, and acknowledging that our paper presents a clear idea, is well explained, tackles an important problem in modeling uncertainty in translation for risk assessment, and recognizing our analysis as thorough.
>
> ### Q: In qualitative comparison, the RGB channels appear misaligned. Images are blurry and low resolution. Presenting FID results.
> We have carefully checked our evaluation and would like to confirm that the RGB channels are not shifted. The color distortions (with clean images as input) in the Facade are caused by the following two reasons. (1) The color information is not encoded in the conditional input (i.e. a segmentation mask). As the segmentation mask only provides structural information, it is infeasible for the model to hallucinate the colors that are exactly in line with the groundtruth images, e.g. the distortions in the channels happen in the high-frequency regions (like inside the window) as it is not encoded in the conditional input. (2) The Facade is a relatively small-scale dataset with 606 samples, while the data covers a wide variety of different colors and shapes. Therefore, color distortions are likely to be present in the output images, as shown in some of the models on Facade. Moreover, this is common in the case where images are downsampled for training to limit the memory footprint. Please note that for the more homogenous (in terms of colors/textures/shape) and slightly larger dataset like Google Maps and MRI such problems do not occur.
>
> Furthermore, in our visualization, half of the images are generated based on noisily perturbed input which could lead to severe blurriness. When adding unseen noisy perturbation at test time, we can observe that many other existing methods are likely to degrade significantly and generate images of poor quality, i.e. severe blurriness and distortions. We adopt such an evaluation protocol with noisy input to test the model’s robustness under unseen perturbation. Nevertheless, even under unseen noisy perturbation, our UGAC still generates images of higher quality that are close to the groundtruth.
>
> As FID does not measure the robustness of the methods across noise levels, we presented other metrics, but FID can help establish the quality of the output at a particular noise level without being affected by minute local distortions. Therefore, below we present FID scores to demonstrate that the fidelity of the generated images is unaffected by minor local distortions. In the following table, we present the FID scores for the Facade datasets. We include all the methods, at all the noise levels (NL0 to NL3), and the final average FID across noise levels. The results are presented for Gaussian distortions.
>
> | Methods | NL0 | NL1 | NL2 | NL3 | Avg. FID |
> |:-----------:|:------:|:-----:|:------:|:----:|:------------:|
> | disGAN | 129.1 |155.7 | 189.6 | 236.1 | 177.6|
> | gcGAN | 134.7 | 178.9 | 211.8 | 257.2 | 195.7|
> | UNIT | 126.2 | 152.5 | 207.4 | 248.6 | 183.7|
> | CUT | 93.8 | 137.6 | 193.3 | 235.8 | 165.1 |
> | Cy.GAN | 100.3 | 148.2 | 210.8 | 253.7 | 178.3 |
> | gCy.GAN | 112.7 | 133.4 | 190.7 | 220.2 | 164.3 |
> | nCy.GAN | 107.5 | 128.3 | 169.6 | 194.3 | 149.9 |
> | UGAC | 95.2 | 99.6 | 107.8 | 114.7 | 104.3|
>
> Here again, we see that UGAC has the lowest average FID score, showing the efficacy of the proposed methods.
>
> ### Q: some changes may be coming from the architectural changes. What are the results of using U-Net and cycle loss without uncertainty modeling?
> Some of our design choices are guided by limited computational resources, such as the need to maintain a small memory footprint, and the method to be sample efficient. UNet [a] and its derivatives like Cascaded UNet [b] have been shown to be sample efficient with lower memory footprint and result in high fidelity images [b]. We use the same generator in all the methods for fairness. Hence, an improved generator architecture may potentially positively effect all the results.
> We would like to re-emphasize that the main contribution of the paper is the uncertainty-aware formulation of cycle consistency in a manner that leads to robustness to noise perturbation. Standard methods degrade faster in the presence of perturbation mainly because of the lack of explicit modeling in the formulation taking care of robustness, and not due to architectural elements. Nevertheless, the following table compares UNet based cycleGAN without uncertainty modeling and UGAC with Gaussian perturbations for the MRI dataset.
>
> | Methods | ASSIM | AMSE | AFID |
> |:-----------:|:------:|:-----:|:------:|
> UNet cycGAN w/o uncertainty |  0.73 | 110.36 | 103.4 |
> UGAC | 0.87 | 68.38 | 87.7 |
>
> Moreover, to demonstrate that the effect is independent of the architectural choice, we present results with a lightweight resnet-derived hourglass-based generator (G1) as described here: https://towardsdatascience.com/using-hourglass-networks-to-understand-human-poses-1e40e349fa15 (modified to output images). The following results show that our method also generalizes well on another generator, with a different architecture.
>
> | Methods | ASSIM | AMSE | AFID |
> |:-----------:|:------:|:-----:|:------:|
> G1 cycGAN w/o uncertainty | 0.71 | 119.53 | 126.3 |
> G1 UGAC | 0.84 | 83.44 | 98.3 |
>
>
> - [a] U-Net: Convolutional Networks for Biomedical Image Segmentation, Olaf Ronneberger, Philipp Fischer, Thomas Brox, MICCAI15
> - [b] MedGAN: Medical image translation using GANs, Karim Armanious et al., Elsevier Computerized Medical Imaging and Graphics, Volume 79, January 2020, 101684
>
> ### Q: Visualizing the top-10 most certain and uncertain examples would be interesting
> Thanks for the nice suggestion. Due to space limitations, we will give visual examples in our supplementary materials. In general, the uncertainty scores are correlated with the errors wrt the groundtruth images. This means that the uncertainty score indicates the generated image quality. Therefore, the most certain/uncertain images are the ones that have relatively higher/poorer visual quality.

---

> > ### Comment · Reviewer_xn61 · 2021-08-10
> > **Misaligned color channels**
> >
> > I appreciate the authors for the thorough feedback! I am in the process of digesting the materials. In the meantime, I just wanted to ask a follow-up question on the seemingly strange color displacement. The reasons the authors mention, that (1) color information is not encoded in the input, and (2) the Facade is a relatively small dataset, are not convincing to me, because the original CycleGAN results (https://taesung.me/cyclegan/2017/03/25/facades.html) don't have such artifacts. Moreover, such artifacts seem very obvious to the discriminator. Is there something I am missing?

---

> > > ### Author Response · Authors · 2021-08-10
> > > **Response to reviewer**
> > >
> > > We thank the reviewer for allowing us to clarify this point within the rebuttal period. We used the same generator architecture for all the methods for a fair comparison, i.e. we adopted the cascaded UNet based generator to ensure sample efficiency with a lower memory footprint. Compared to the generator in the vanilla CycleGAN, UNet generator does not model the color distribution so well when using a small dataset, e.g. Facade with 600 images, thus causing the color distortion. However, we do not observe this on larger datasets, e.g. the Google Maps (Figure 4). We observed that the original CycleGAN generator is good at modeling color distribution even with small datasets, but in addition to requiring a higher computational effort, it can not handle unseen noisy perturbations at test time which is our focus in this work.
> > >
> > > In the table below, we show that the original CycleGAN generator (Cy.GAN-vanilla, checkpoint taken from http://efrosgans.eecs.berkeley.edu/cyclegan/pretrained_models/) performs worse under noisy perturbations, i.e. "Cy.GAN - Vanilla"  (and also "Cy.GAN - UNet") degrade significantly under noisy input at perturbation levels NL1/NL2/NL3.
> > > As "Cy.GAN - vanilla" has a high-capacity generator, the results for clean input images (at NL0) are quantitatively and qualitatively better (we observe no color distortions), but its performance is consistently worse under varying noisy perturbation levels. This again confirms that the degradation in output quality due to perturbed input (i.e., robustness towards perturbation) is not a consequence of architectural elements but the underlying learning scheme utilized (in case of “Cy.GAN - vanilla”, L1 loss function along with the discriminator support was used for training.)
> > >
> > > We also observe that at the noise level NL0, “Cy.GAN - vanilla” is the best in terms of FID whereas starting from NL1, our UGAC method outperforms it. Furthermore, at NL2 and NL3, "Cy.GAN - vanilla" obtains FID scores higher than the "Cy.GAN - UNet" and UGAC (i.e., “Cy.GAN - vanilla” is worse than “Cy.GAN-UNet” and UGAC).
> > >
> > > | Methods | NL0 | NL1 | NL2 | NL3 | Avg.FID (AFID) |
> > > |:-----------:|:------:|:------:|:-----:|:------:|:--------------------|
> > > Cy.GAN - UNet | 100.3 | 148.2 | 210.8 | 253.7 | 178.3 |
> > > Cy.GAN - vanilla | 81.7 | 133.6 | 229.3 | 286.9 | 182.9 |
> > > UGAC - UNet | 95.2 | 99.6 | 107.8 | 114.7 | 104.3 |
> > >
> > > In summary, although our UNet based generator may not model colors well with small data regimes, unlike CycleGAN, it is robust under varying noise levels and it is efficient. Designing a generator architecture on such regimes is not a trivial issue and an interesting research direction, but it is not our main focus. Therefore, when it comes to modeling robustness under unseen noise perturbations, our uncertainty-aware cycle consistency loss works better than others. We will add this discussion in our revised version.

---

### Official Review · Reviewer_e4fm · 2021-07-21

**Rating:** 7
**Confidence:** 2

**Summary:**

This paper introduces a new method for unpaired image to image translation that addresses uncertainty estimation and robustness. The method is based on an adaptive loss function that accounts for spatially-varying noise levels. This loss function motivated by Bayesian modelling and it is incorporated in a cycle-consistent loss. The experimental setting consists in training the proposed method and other competitors on clean data and test it on noisy data. The method is evaluated on different dataset including street scene images and medical images. According to a measure of performance on noisy data, the method is shown to outperform existing methods. Last, the method outputs uncertainty maps which are the parameters of the spatially adaptive loss function, these are shown to positively correlate with the residual losses.

**Limitations And Societal Impact:**

The limitations of the work is absent from the conclusion. It is of importance that the authors discuss the potential pitfalls of the method.

**Main Review:**

The motivation of the paper is clear and is of definite interest. Quantification of uncertainty matters for practical applications. At runtime, uncertainty comes from at least two sources: the uncertainty on the learned model parameters and the uncertainty in the input. This paper incorporates the uncertainty of the input directly in the loss, via a weighted $L^p$ loss in which $p$ and the weight are pixel dependent. The actual form is motivated by a Bayesian analysis. These loss parameters are learnt and depend on the entire image.
The paper is well written and easy to read.


Questions:
- Figure 3 shows that the method outperforms the other methods even at $0$ noise level. Since the method is made to take learn noise in the loss function, it is not intuitive why a performance improvement can be achieved at $0$ noise level. Can the authors elaborate on this?
- Data augmentation can be used to gain robustness with respect to noise in the data. How would it compare with the proposed method?
- In the situation where the parameters of the data noise is known, it is rather unclear that the model would be able to retrieve these parameters. Would it be possible to test the model in this situation? My guess is that the found parameters will be very dependent on the image and will not recover the true underlying parameters. As a conclusion, it is non easily interpretable what is learnt in this loss function.



Typos:
- page 4 : equation 9 and 5 a dot is missing
- "boarder impact"




**Time Spent Reviewing:**

3

---

> ### Author Response · Authors · 2021-08-09
> **Response to reviewer**
>
> We sincerely thank the reviewer for acknowledging that our paper is well-written, easy to read while tackling the practical problem of estimating uncertainty with predictions. We will fix the typos and address the further comments below.
>
> ### Q: Can UGAC model parameters of the underlying noise distribution if it is known?
> To elaborate the reviewer’s comment with an example, say the data noise is coming from a Gaussian distribution with $0$ mean and variance of $0.6$, i.e., $\mathcal{N}(0, 0.6)$. In such a situation, given enough training data, our model with Generalized Gaussian Distribution (see Eq. (3) in Section 3.2) is capable of learning to predict the $\alpha$ close to $\sqrt{0.6 * \frac{\Gamma(1/2)}{\Gamma(3/2)}}$ and $\beta$ close to $2$, i,e., a residual distribution equivalent to $GGD(0, \sqrt{0.6 * \frac{\Gamma(1/2)}{\Gamma(3/2)}} , 2) \equiv \mathcal{N}(0, 0.6)$. This is because variance for GGD is given by, $\sigma^2 = \alpha^2\frac{\Gamma(3/\beta)}{\Gamma(1/\beta)}$.
>
> To further extend on this argument, we analyze the case where the underlying noise is some complex parametrized distribution and not Gaussian as in the example above -- Even in this case, the model tries to approximate the underlying distribution with the generalized Gaussian distribution (GGD) which varies per pixel.
> Note that this assumption is much more liberal than both homo/hetero-scedastic Gaussian/Laplacian distribution because it is able to capture all the heavier/lighter-tailed distributions (along with all the possible Gaussian/Laplacian distributions) that are beyond the modeling capabilities of Gaussian/Laplacian alone.
>
> ### Q: How is data augmentation compared with UGAC?
> UGAC is orthogonal to data augmentation. While data augmentation operates in the input data space, our UGAC incorporates an uncertainty model component in the output space. As their regularization effects are introduced at different spaces, combining UGAC with data augmentation may even further improve model generalization.
> Furthermore, UGAC can especially enhance model robustness towards unseen noisy perturbation in input data. To ensure model robustness, data augmentation has to cover all types of possible noisy variations in the input, which is infeasible as the model may encounter unseen noisy perturbation at test time. Thus, even though various data augmentation strategies (such as cropping, color jittering, or adding noises) generally improve the performance of many methods, they do not guarantee model robustness towards unseen noisy perturbation.
> In contrast, UGAC is capable of performing well under different types of unseen noises, across various unseen noise levels, as shown in our experiments.
>
> Finally, UGAC has unique merit for modeling uncertainty in the output (that is not possible with data augmentation alone), which serves as an informative indicator of the generated image quality (e.g. Figure 7 shows the correlation between the estimated uncertainty scores indicates the underlying errors wrt the groundtruth).
>
> ### Q: Superior performance at NL0 with GGD?
> In the absence of noise in the input images, $\mathcal{L}_1$ or $\mathcal{L}_2$ norm-based loss functions (that are typically used in existing methods) may not be optimal. As we discussed in Section 3, the optimal norm (i.e., the value of $p$ in $\mathcal{L}_p$ ) may be different from 1 or 2. Moreover, the optimal $p$ may also vary from pixel to pixel. Fixed norms like $\mathcal{L}_1$ and $\mathcal{L}_2$ do not model this behavior as a result may not lead to optimal performance.

---

### Decision · Program_Chairs · 2021-09-27

**Decision:**

Accept (Poster)

**Comment:**

After discussion, the reviewers are all for accepting the work. It is well-motivated and presents its ideas well. There are some concerns regarding image artifacts (particularly, the color channels being distorted). The rebuttal does not really address this concern as no prior work which use very similar architectures suffer from this phenomena. I highly recommend the authors look into resolving or clarifying why this happens, and incorporating this discussion in their paper.